# Application of Mode-Adaptive Bidirectional Pushover Analysis to an Irregular Reinforced Concrete Building Retrofitted via Base Isolation

**Kenji Fujii** [1,*] and **Takumi Masuda** [2]

1   Department of Architecture, Faculty of Creative Engineering, Chiba Institute of Technology, Chiba 275-0016, Japan
2   Graduate School of Creative Engineering, Chiba Institute of Technology, Chiba 275-0016, Japan; s16b1147bq@s.chibakoudai.jp
*   Correspondence: kenji.fujii@p.chibakoudai.jp

**Featured Application: Seismic response evaluation of a base-isolated buildings: Seismic rehabilitation design for reinforced concrete buildings using the base-isolation technique.**

**Abstract:** In this article, the applicability of mode-adaptive bidirectional pushover analysis (MABPA) to base-isolated irregular buildings was evaluated. The point of the updated MABPA is that the peaks of the first and second modal responses are predicted considering the energy balance during a half cycle of the structural response. In the numerical examples, the main building of the former Uto City Hall, which was severely damaged in the 2016 Kumamoto earthquake, was investigated as a case study for the retrofitting of an irregular reinforced concrete building using the base-isolation technique. The comparisons between the predicted peak response by MABPA and nonlinear time-history analysis results showed that the peak relative displacement can be properly predicted by MABPA. The results also showed that the performance of the retrofitted building models was satisfactory for the ground motion considered in this study, including the recorded motions in the 2016 Kumamoto earthquake.

**Keywords:** seismic isolation; asymmetric building; mode-adaptive bidirectional pushover analysis (MABPA); seismic retrofit; momentary energy input

## 1. Introduction

### 1.1. Background

Seismic isolation is widely applied to buildings for earthquake protection in earthquake-prone countries [1]. Unlike in the case of traditional (non-isolated) earthquake-resistant structures, seismic isolation ensures that the behaviors of building structures are within the elastic range and reduces acceleration in buildings during large earthquakes [2]. Therefore, this technique is applied to not only new buildings, but also existing reinforced concrete and masonry buildings, including historical structures [3–14]. The isolation layer consists of isolators and dampers. Isolators support the gravity loads of the superstructure. The horizontal flexibility and centering capability of seismically isolated buildings is also achieved by isolators [1]. Dampers provide the energy dissipation capacity for reducing the seismic responses. Although it is very common to use hysteresis dampers (e.g., steel dampers) and/or oil dampers in the isolation layer, there are several studies about the innovative seismic isolation systems in recent years, e.g., [15–22].

In general, there is some degree of irregularity in building structures, whether they are newly designed or preexisting. Therefore, the problem of torsional response due to plan irregularities needs to be studied for base-isolated structures, as well as non-isolated structures. Several researchers have investigated the seismic behavior of base-isolated

buildings with plan irregularities [23–36]; these include fundamental parametric studies using idealized models [23–27], as well as studies using more realistic frame building models [28–36]. Most of these studies are based on nonlinear time-history analyses [23–35]. However, there are a few investigations that examine the applicability of nonlinear static procedures to base-isolated buildings with asymmetry. Kilar and Koren [36] examined the applicability of the extended N2 method [37,38], which is one of the variants of the nonlinear static procedures, to the base-isolated asymmetric buildings. They found the nonlinear peak response of base-isolated asymmetric building can be predicted by the extended N2 method. However, they examined only those whose superstructure was regular in elevation. Therefore, further investigations are needed for the applicability of the nonlinear static procedures, especially for base-isolated building with plan and elevation irregularities.

*1.2. Motivation*

There were two main motivations for this study. The first was to extend nonlinear static analyses to base-isolated buildings with irregularities, and the second was to predict the nonlinear peak responses of base-isolated buildings according to the concept of energy input.

With respect to the first motivation, the authors proposed mode-adaptive bi-directional pushover analysis (MABPA) [39–44]. This is a variant of nonlinear static analysis and was originally proposed for the seismic analysis of non-isolated asymmetric buildings subjected to horizontal bidirectional excitation. The first version of MABPA was proposed for non-isolated asymmetric buildings with regular elevation [39]; it was then updated following the development of displacement-based mode-adaptive pushover (DB-MAP) analysis [40]. This version has been applied to reinforced concrete asymmetric frame buildings with buckling-restrained braces [41] and to building models with bidirectional setbacks [42]. The applicability of MABPA has been discussed and evaluated based on the effective modal mass ratio of the first two modes [43]. The seismic capacity of an existing irregular building severely damaged in the 2016 Kumamoto earthquake (the former Uto City Hall) has been evaluated using MABPA [44]. Looking back on the development of MABPA, the logical next step should be to extend the method to base-isolated buildings with irregularities. Considering the case in which the seismic isolation period is well separated from the natural period of a superstructure, the behavior of the superstructure may be that of a rigid body, as discussed by several researchers, e.g., [27]. In such cases, the effective modal mass ratio of the first and second modes is close to unity, provided that the torsional resistance at the isolation layer is sufficient. It is expected that the seismic response of such a base-isolated building with plan irregularities will be accurately predicted by MABPA.

In the nonlinear static analysis shown in the American Society of Civil Engineers ASCE/SEI 41-17 document [3], the reduced spectrum considering the effective damping is used to predict the target displacement for a nonlinear static analysis. Similarly, as shown in Notification No. 2009 of the Ministry of Construction of Japan [45], the equivalent linearization technique can be used for target displacement evaluations of base-isolated buildings. However, several researchers have examined the responses of long-period building structures subjected to pulse-like ground motions, e.g., Mazza examined a base-isolated building [29,31] and Güneş and Ulucan studied a tall reinforced concrete building [46]. Because the effective damping is calculated based on a steady response having the same displacement amplitude in the positive and negative directions, the use of effective damping for the prediction of the peak responses of base-isolated structures subjected to pulse-like ground motion is questionable. From this viewpoint, an alternative concept for predicting the peak response is required. Accordingly, the second motivation of this study was to investigate the use of the energy concept.

The concept of energy input was introduced by Akiyama in the 1980 s [47] and is implemented in the design recommendation for seismically isolated buildings presented by the Architectural Institute of Japan [2]. In Akiyama's theory, the *total input energy* is a suit-

able seismic intensity parameter to access the cumulative response of a structure. Instead of the total input energy, Inoue et al. proposed the *maximum momentary input energy* [48–50] as an intensity parameter related to peak displacement; nonlinear peak displacement can be predicted by equating the maximum momentary input energy and the cumulative hysteresis energy during a half cycle of the structural response. Following the work of Inoue et al., the authors investigated the relationship between the maximum momentary input energy and the total input energy of an elastic single-degree-of-freedom (SDOF) model [51]. In addition, the concept of the momentary input energy was extended to consider bidirectional horizontal excitation [52,53]. Specifically, the time-varying function of the momentary energy input has been formulated for unidirectional [51] and bidirectional [52] excitation: This function can be calculated from the transfer function of the model and the complex Fourier coefficients of the ground acceleration. Using the time-varying function, both energy parameters, the total and maximum momentary input energy, can be accurately calculated. In a previous study [53], it was shown that the nonlinear peak displacement and the cumulative energy of the isotropic two-degree-of-freedom model representing a reinforced concrete building can be properly predicted using a time-varying function. Therefore, the use of a time-varying function of the momentary energy input is promising for seismic response predictions for base-isolated buildings.

### 1.3. Objectives

Based on the above discussion, the following questions were addressed in this paper.

- Is MABPA capable of predicting the peak response of irregular base-isolated buildings?
- The prediction of the peak equivalent displacement of the first two modal responses is an essential step in MABPA. For this, the relationship between the maximum momentary input energy and the peak displacement needs to be properly evaluated. How can this relationship be evaluated from the pushover analysis results?
- In the prediction of the maximum momentary input energy of the first two modal responses, the effect of simultaneous bidirectional excitation needs to be considered. Can the upper bound of the peak equivalent displacement of the first two modal responses be predicted using the bidirectional maximum momentary input energy spectrum [52]?

In this study, the applicability of mode-adaptive bidirectional pushover analysis (MABPA) to base-isolated irregular buildings was evaluated. The point of the updated MABPA is that the peaks of the first and second modal responses are predicted considering the energy balance during a half cycle of the structural response. In the numerical examples, the main building of the former Uto City Hall [44], which was severely damaged in the 2016 Kumamoto earthquake, was investigated as a case study for the retrofitting of an irregular reinforced concrete building using the base-isolation technique. The nonlinear peak responses of two retrofitted building models subjected to bidirectional excitation were investigated via a time-history analysis using artificial and recorded ground motion datasets. Then, their peak responses were predicted by MABPA to evaluate its accuracy. Note that of the many types of dampers used nowadays in base-isolated buildings, only hysteresis dampers were considered in this study; this is because they are easily implemented in nonlinear static analysis. The applicability of MABPA to base-isolated buildings with other types of dampers, such as oil and tuned viscous mass dampers [18], will be the next phase of this study.

The rest of paper is organized as follows. Section 2 presents an outline of MABPA, followed by the prediction procedure of the maximum equivalent displacement conducted using the maximum momentary input energy. Section 3 briefly presents information concerning the original building. Then, two retrofitted building models using the base-isolation technique are presented, as well as the ground motion data used in the nonlinear time-history analysis. The validation of the prediction of the peak response is discussed in Section 4. Discussions focused on (i) the relationship between the maximum equivalent displacement and the maximum momentary input energy, (ii) the predictability of

the largest maximum momentary input energy from the bidirectional momentary input energy spectrum, and (iii) the accuracy of the upper bound of the maximum equivalent displacement of the first two modes are presented in Section 5. The conclusions and future directions of the study are discussed in Section 6.

## 2. Description of MABPA

### 2.1. Outline of MABPA

Figure 1 shows the flow of the updated MABPA. Here, the U-axis is the principal axis of the first modal response, while the V-axis is the axis perpendicular to the U-axis, following previous studies [39,41]. In this study, the predictions of the peak responses of the first and second modes (steps 2 and 4) were updated as follows:

- The bidirectional momentary input energy proposed in the previous study was applied as the seismic intensity parameter.
- The peak response of each mode was predicted from the energy balance in a half cycle of the structural response.

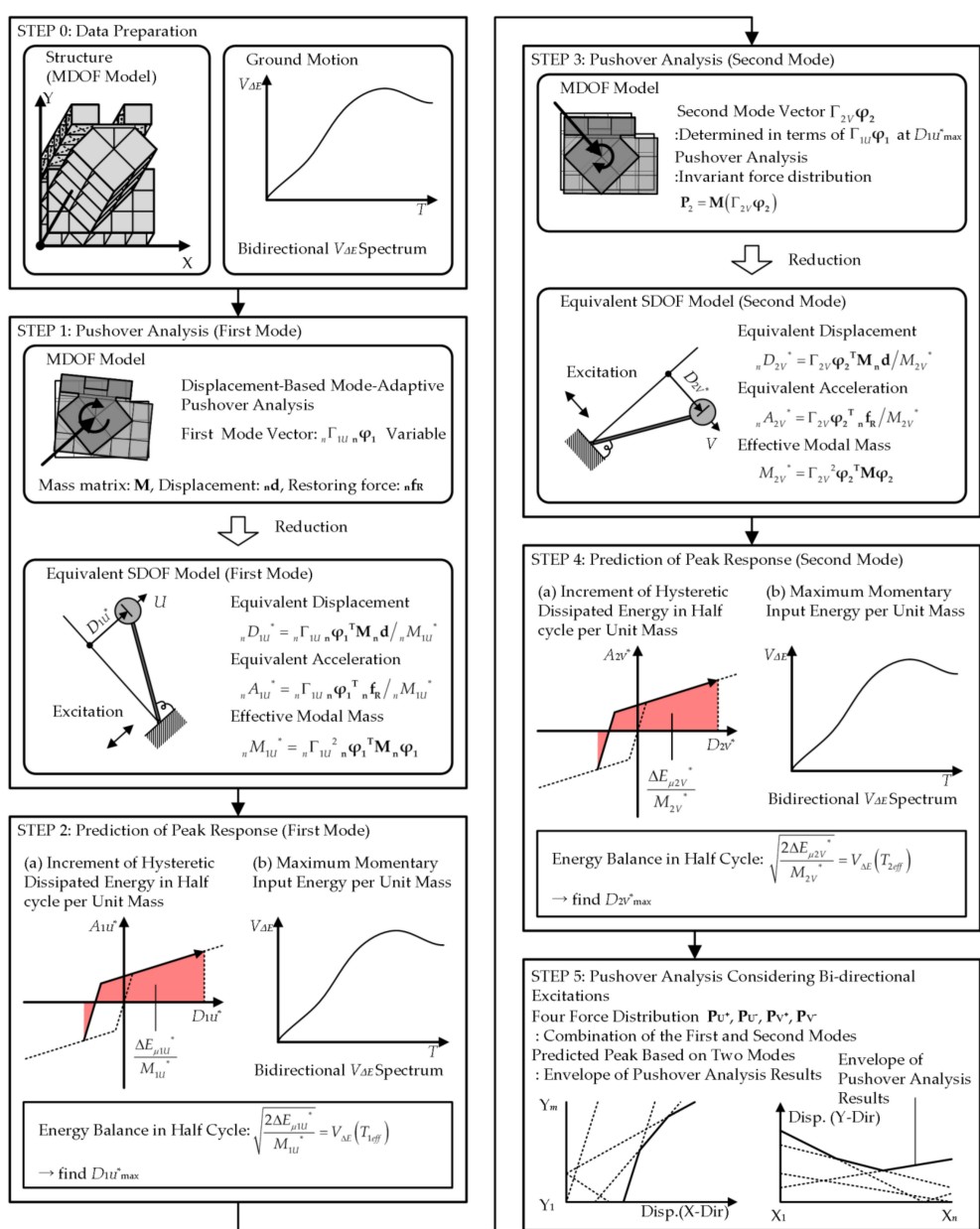

**Figure 1.** Flow of updated mode-adaptive bi-directional pushover analysis (MABPA).

It was assumed that the upper bound of the unidirectional maximum momentary input energy considering the various directions of the seismic input is approximated by the bidirectional maximum momentary input energy. For the prediction of the peak equivalent displacements of the first and second modal responses, the bidirectional maximum momentary input energy spectrum (bidirectional $V_{\Delta E}$ spectrum) was used. This is because the influences of the simultaneous input of two horizontal components are included in the bidirectional $V_{\Delta E}$ spectrum, as discussed in previous studies [52,53]. Note that the maximum momentary input energy of each modal response varies depending on the angle of incidence of the horizontal ground motion. The assumption that the maximum momentary input energy of each modal response can be predicted from the bidirectional $V_{\Delta E}$ spectrum can lead to conservative predictions. The validation of this assumption is discussed in Section 5.

### 2.2. Prediction of the Peak Response using the Momentary Energy Input

2.2.1. Calculation of the Bidirectional Momentary Input Energy Spectrum

The bidirectional momentary input energy spectrum, which was proposed in previous studies [52,53], can be calculated as follows.

First, the complex Fourier coefficient of the two orthogonal components of horizontal ground acceleration, $c_{\xi,n}$ and $c_{\zeta,n}$, can be calculated via the discrete Fourier transform of the two horizontal ground acceleration components $a_{g\xi}(t)$ and $a_{g\zeta}(t)$, respectively.

The displacement and velocity transfer function of the linear system with viscous and complex damping can be calculated as follows:

$$H_{CVD}(i\omega_n) = \frac{1}{\omega_0{}^2 - \omega_n{}^2 + 2\omega_0\{h\omega_n + \beta\omega_0 \mathrm{sgn}(\omega_n)\}i}, \quad H_{CVV}(i\omega_n) = i\omega_n H_{CVD}(i\omega_n). \tag{1}$$

In Equation (1), $\omega_0 (= 2\pi/T)$ is the natural circular frequency, $h$ and $\beta$ are the viscous and complex damping of the linear system, respectively; $\omega_n$ is the natural frequency of the $n$th harmonic. From the complex Fourier coefficients ($c_{\xi,n}$ and $c_{\zeta,n}$) and the transfer functions of the linear system ($H_{CVD}(i\omega_n)$ and $H_{CVV}(i\omega_n)$), the duration of a half cycle of response ($\Delta t$) can be calculated as:

$$\Delta t = \pi \sqrt{\sum_{n=1}^{N} |H_{CVD}(i\omega_n)|^2 \left\{ |c_{\xi,n}|^2 + |c_{\zeta,n}|^2 \right\} \Big/ \sum_{n=1}^{N} |H_{CVV}(i\omega_n)|^2 \left\{ |c_{\xi,n}|^2 + |c_{\zeta,n}|^2 \right\}}. \tag{2}$$

Then, the Fourier coefficient of the time-varying function of the momentary energy input can be calculated as follows:

$$E_{\Delta BI,n}{}^* = \begin{cases} \frac{\sin(\omega_n \Delta t/2)}{\omega_n \Delta t/2} \sum_{n_1=n+1}^{N} \left\{ H_V(i\omega_{n_1}) + H_V(-i\omega_{n_1-n}) \right\} \left\{ c_{\xi,n_1} c_{\xi,-(n_1-n)} + c_{\zeta,n_1} c_{\zeta,-(n_1-n)} \right\} & : n > 0 \\ 2 \sum_{n_1=1}^{N} \mathrm{Re}\{H_V(i\omega_{n_1})\} \left\{ |c_{\xi,n_1}|^2 + |c_{\zeta,n_1}|^2 \right\} & : n = 0 \\ \overline{E_{\Delta BI,n}{}^*} & : n < 0 \end{cases} \tag{3}$$

The momentary input energy per unit mass at time $t$ can be calculated as follows:

$$\frac{\Delta E_{BI}(t)}{m} = \int_{t-\Delta t/2}^{t+\Delta t/2} \sum_{n=-N+1}^{N-1} E_{\Delta BI,n}{}^* \exp(i\omega_n t) dt. \tag{4}$$

The maximum momentary input energy per unit mass ($\Delta E_{BI,\max}/m$) can be evaluated as the maximum value calculated by Equation (4) over the course of the seismic event.

The bidirectional maximum momentary input energy spectrum (the bidirectional $V_{\Delta E}$ spectrum) can be calculated as:

$$V_{\Delta E} = \sqrt{2\Delta E_{BI,\max}/m}. \tag{5}$$

The bidirectional $V_{\Delta E}$ spectrum is prepared at the beginning of the MABPA (step 0). The range of the natural period ($T$) needs to be properly considered. In this study, a range of $T$ from 0.1 to 6.0 s (longer than the seismic isolation period) was considered. The assumption of damping is also important. Because only a hysteresis (displacement-dependent) damper was installed in the isolation layer, complex damping was chosen in this study. The complex damping ratios ($\beta$) were set to 0.10, 0.20, and 0.30.

### 2.2.2. Formulation of the Effective Period and the Hysteretic Dissipated Energy in a Half Cycle

To predict the peak responses of the first two modes, bilinear idealization of the equivalent acceleration-equivalent displacement relationship was made as follows. In the following discussion, the formulations are made for the first modal response. Figure 2 shows the bilinear idealization procedure used in this study.

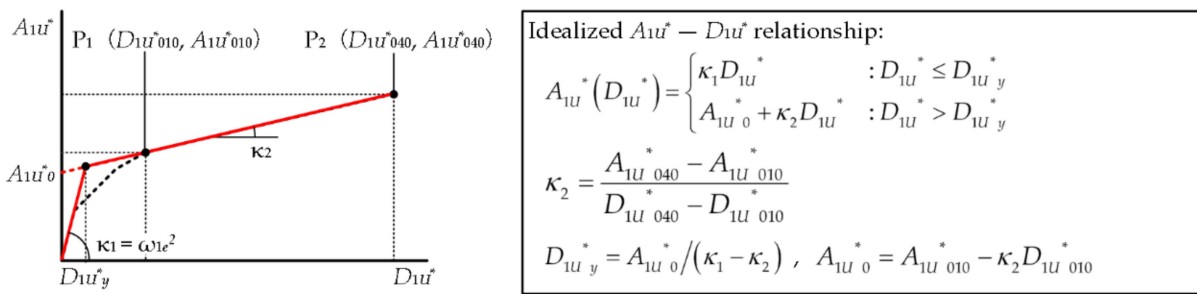

**Figure 2.** Bilinear idealization of the equivalent acceleration-equivalent displacement relationship.

Then, the effective period ($T_{1eff}$) and the hysteretic dissipated energy in the half cycle per unit mass ($\Delta E_{\mu 1U}{}^{*}/M_{1U}{}^{*}$) were formulated. Figure 3 shows the modeling of the structural response in a half cycle used to predict the peak response. In Figure 3, the parameter $\eta$ is the displacement ratio in the positive and negative directions and was assumed to be in the range from 0 to 1.

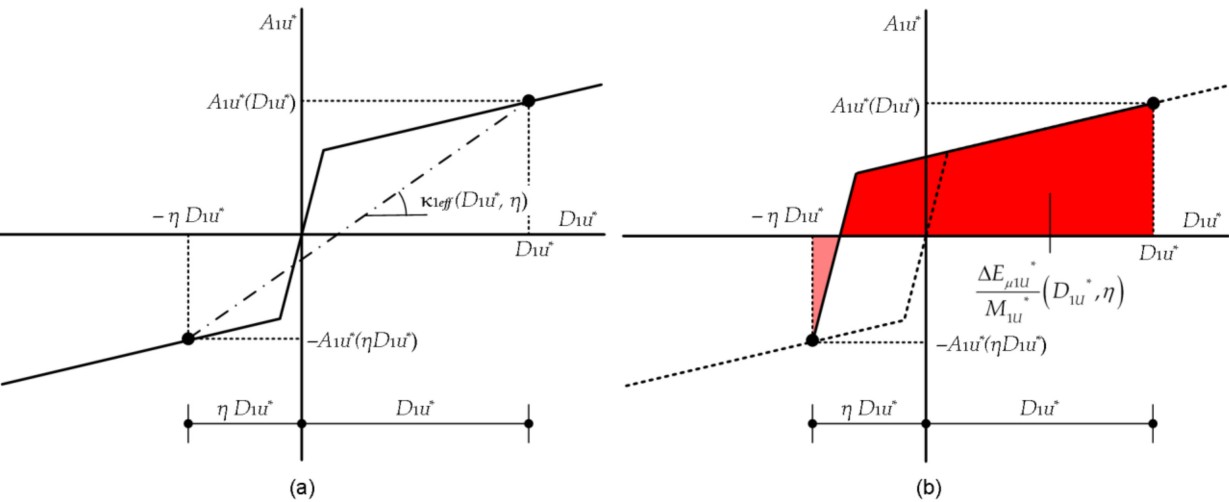

**Figure 3.** Modeling of the structural response in a half cycle used to predict the peak response: (**a**) The effective slope ($\kappa_{1eff}$) and (**b**) the hysteretic dissipated energy in a half cycle per unit mass ($\Delta E_{\mu 1U}/M_{1U}{}^{*}$).

In this study, the effective period of the first mode corresponding to $D_{1U}{}^*$ was calculated as:

$$T_{1eff}(D_{1U}{}^*) = 2\pi / \sqrt{\kappa_{1eff}(D_{1U}{}^*)}. \tag{6}$$

In addition, the equivalent velocity of the hysteretic dissipated energy in a half cycle corresponding to $D_{1U}{}^*$ was calculated as:

$$V_{\Delta E\mu1U}{}^*(D_{1U}{}^*) = \sqrt{\frac{2\Delta E_{\mu1U}{}^*}{M_{1U}{}^*}(D_{1U}{}^*)}. \tag{7}$$

The effective slope corresponding to $D_{1U}{}^*$ was calculated as:

$$\kappa_{1eff}(D_{1U}{}^*) = \int_0^1 \kappa_{1eff}(D_{1U}{}^*,\eta)d\eta$$
$$= \begin{cases} \kappa_1 & : D_{1U}{}^* \le D_{1U}{}^*{}_y \\ (2\ln 2)\frac{A_{1U}{}^*{}_0}{D_{1U}{}^*} + \kappa_2 - \left\{\ln\left(1 + \frac{D_{1U}{}^*{}_y}{D_{1U}{}^*}\right)\right\}\left(\frac{A_{1U}{}^*{}_0}{D_{1U}{}^*} + \kappa_1 - \kappa_2\right) + \frac{D_{1U}{}^*{}_y}{D_{1U}{}^*}(\kappa_1 - \kappa_2) & : D_{1U}{}^* > D_{1U}{}^*{}_y \end{cases} \tag{8}$$

The hysteretic dissipated energy in a half cycle per unit mass corresponding to $D_{1U}{}^*$ was calculated as:

$$\frac{\Delta E_{\mu1U}{}^*}{M_{1U}{}^*}(D_{1U}{}^*) = \int_0^1 \frac{\Delta E_{\mu1U}{}^*}{M_{1U}{}^*}(D_{1U}{}^*,\eta)d\eta$$
$$= \begin{cases} \frac{1}{3}\kappa_1 D_{1U}{}^{*2} & : D_{1U}{}^* \le D_{1U}{}^*{}_y \\ \frac{1}{3}\kappa_2 D_{1U}{}^{*2} + \frac{3}{2}A_{1U}{}^*{}_0 D_{1U}{}^*\left\{\left(1 - \frac{D_{1U}{}^*{}_y}{D_{1U}{}^*}\right)^2 + \frac{2}{3}\frac{D_{1U}{}^*{}_y}{D_{1U}{}^*}\left(1 - \frac{2}{3}\frac{D_{1U}{}^*{}_y}{D_{1U}{}^*}\right)\right\} & : D_{1U}{}^* > D_{1U}{}^*{}_y \end{cases} \tag{9}$$

The predicted peak response $D_{1U}{}^*{}_{max}$ was obtained from the following equation:

$$V_{\Delta E\mu1U}{}^*\left\{T_{1eff}(D_{1U}{}^*{}_{max})\right\} = V_{\Delta E}\left\{T_{1eff}(D_{1U}{}^*{}_{max})\right\}. \tag{10}$$

Note that no viscous damping was considered because the isolation layer was assumed to have no viscous damping.

## 3. Description of the Retrofitted Building Models and Ground Motion Datasets

### 3.1. Original Building

Figure 4 shows a simplified structural plan and the elevation of the main building of the former Uto City Hall [44]. This five-story reinforced concrete irregular building was constructed in 1965.

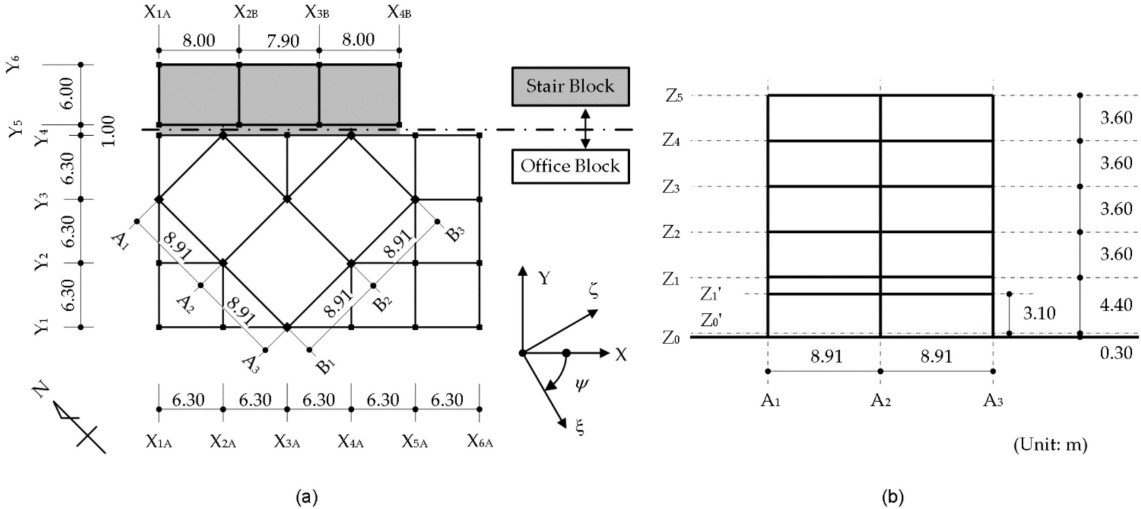

**Figure 4.** Simplified structural plan and elevation of the main building of the former Uto City Hall [44]: (**a**) Structural plan (Level $Z_0$) and (**b**) simplified plan elevation (frame $B_1$).

Figure 5 shows the former Uto City Hall after the 2016 Kumamoto earthquake. After the 2016 Kumamoto earthquake, this building was demolished. Details concerning this building can be found in the literature [44].

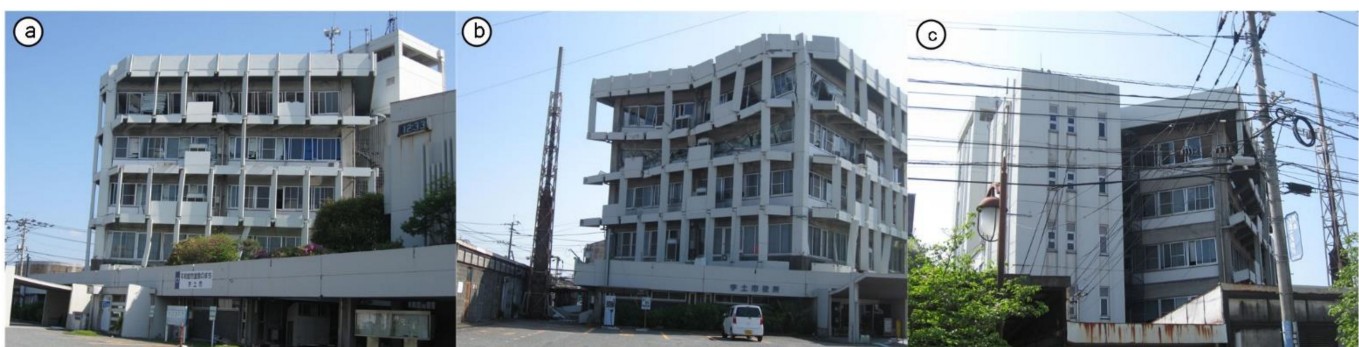

**Figure 5.** View of the former Uto City Hall after the 2016 Kumamoto earthquake [44]. Photographs were taken from (**a**) the south, (**b**) the southwest, and (**c**) the north.

Figure 6 shows the soil properties at the former K-NET Uto station, which was within the site of the Uto City Hall at the time of the 2016 Kumamoto earthquake. Note that those soil properties at the former K-NET Uto station are available from the K-NET website [54].

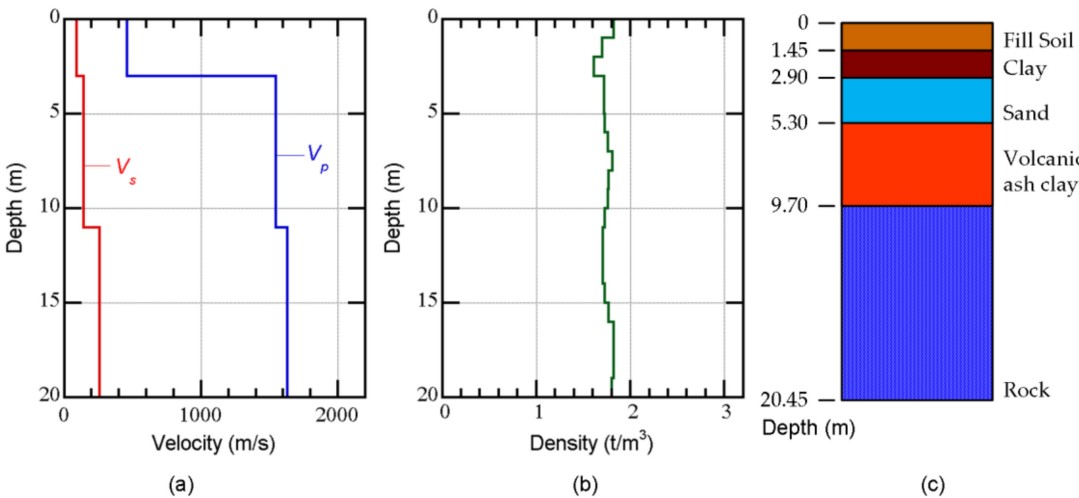

**Figure 6.** Soil profiles of the former K-NET Uto station. Figures were made from the data provided by the National Research Institute for Earth Science and Disaster Resilience (NIED). (**a**) Primary and shear wave profile ($V_p$: P-wave velocity, $V_s$: S-wave velocity); (**b**) density profile; (**c**) soil column.

### 3.2. Properties of the Isolation Layer

Two retrofitted building models were examined in this study to consider different seismic isolation periods ($T_f$). The properties of the isolation layer were determined as follows, based on the energy-balanced design method introduced in the design recommendation [2].

First, the mass and the moment of inertia of each floor were calculated. Table 1 shows the floor mass ($m_j$), moment of inertia ($I_j$), and the radius of gyration of the floor mass ($r_j$) for the *j*th floor. Here, the values of the floor mass and the moment of inertia above level 1 were taken from a previous study [44], while the weight per unit area at level 0 was assumed to be 24 kN/m$^2$ for the calculation of the floor mass and the moment of inertia of each floor. Therefore, the calculated total mass ($M$) was 5412 t.

**Table 1.** Floor mass, moment of inertia, and radius of gyration of the floor mass of each floor level.

| Floor Level $j$ | Floor Mass $m_j$ (t) | Moment of Inertia $I_j$ ($\times 10^3$ tm$^2$) | Radius of Gyration of Floor Mass $r_j$ (m) |
|---|---|---|---|
| 5 | 677.8 | 78.37 | 10.75 |
| 4 | 548.5 | 62.85 | 10.70 |
| 3 | 543.0 | 62.50 | 10.73 |
| 2 | 581.0 | 67.12 | 10.75 |
| 1 | 1208.1 | 199.0 | 12.83 |
| 0 | 1853.7 | 274.9 | 12.18 |

Figure 7 shows the horizontal response spectra of the recorded ground motions of the first (14 April 2016: UTO0414) and second (16 April 2016: UTO0416) earthquakes observed at the K-NET Uto station [54]. Note that the spectral acceleration ($S_A$) and velocity ($S_V$) shown in Figure 7a,b were calculated as the absolute (vector) value of the two horizontal components. The total input energy spectrum (the $V_I$ spectrum) was also calculated considering the simultaneous input of the two horizontal components. The viscous damping ratio was set to 0.05 for the calculations of $S_A$ and $S_V$, while it was set to 0.10 for the calculation of $V_I$ following the work of Akiyama [47].

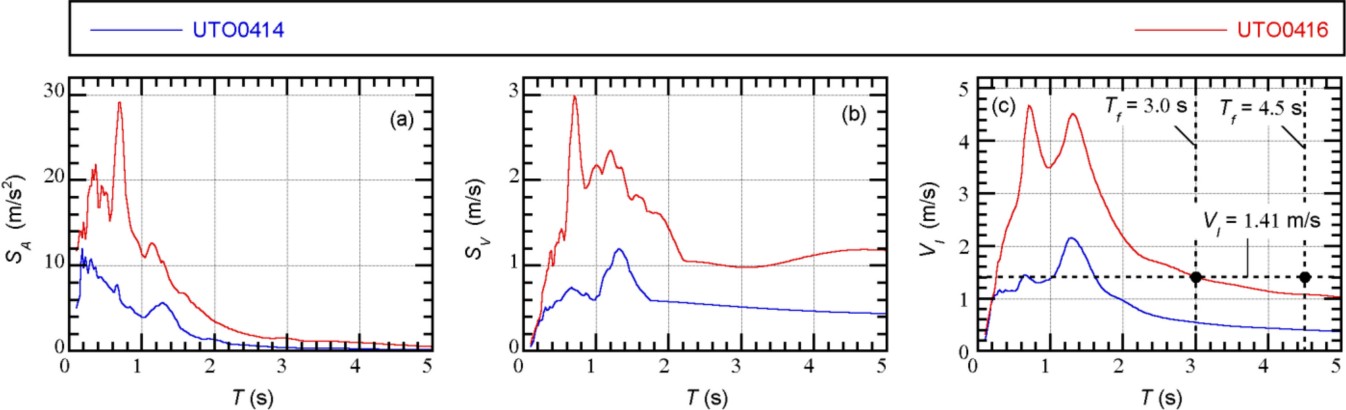

**Figure 7.** Horizontal response spectra of the recorded ground motions at the K-NET Uto station: (**a**) Elastic acceleration response spectrum (damping ratio: 0.05); (**b**) elastic velocity response spectrum (damping ratio: 0.05); (**c**) elastic total input energy spectrum (damping ratio: 0.10).

In this study, the range of the seismic isolation period ($T_f$) was considered to be from 3 to 4.5 s when determining the properties of the isolation layer. Based on Figure 7c, the equivalent velocity of the total input energy ($V_I$) was set to 1.41 m/s to determine the properties of the isolation layer.

Assuming that the superstructure is a rigid body, the equation of the energy balance is expressed as:

$$\frac{1}{2}K_f\delta_{\max}{}^2 + 4\,{}_sn_sQ_y\delta_{\max} = \frac{1}{2}MV_I{}^2. \tag{11}$$

In Equation (11), $\delta_{\max}$ is the maximum horizontal displacement, $_sn$ is the equivalent number of repetitions, $K_f$ is the total horizontal stiffness of the flexible element, and $_sQ_y$ is the total yield strength of the rigid plastic element. Equation (11) can be rewritten as:

$$\left(\frac{2\pi}{T_f}\right)^2\delta_{\max}{}^2 + 8\,{}_sn_s\alpha_yg\delta_{\max} = V_I{}^2, \tag{12}$$

$$\text{where } K_f = M\left(\frac{2\pi}{T_f}\right)^2, {}_s\alpha_y = \frac{{}_sQ_y}{Mg}. \tag{13}$$

In Equation (13), $g$ is the acceleration due to gravity, assumed to be 9.8 m/s$^2$, and $_s\alpha_y$ is the yielding shear strength coefficient. In this study, the design-allowable horizontal displacement ($\delta_a$) was set to 0.40 m, while the value of $_s n$ was set to 2 following the design recommendation [2]. Then, the two parameters of the isolation layer, $T_f$ and $_s\alpha_y$, were adjusted, such that the following condition was satisfied:

$$\left(\frac{2\pi}{T_f}\right)^2 \delta_a{}^2 + 16\,_s\alpha_y g \delta_a \geq V_I{}^2. \tag{14}$$

The isolation layer below level 0 comprises natural rubber bearings (NRBs), elastic sliding bearings (ESBs), and steel dampers. Figure 8 shows the layout of the isolators and dampers in the isolation layer of the two retrofitted building models: Model-Tf34, shown in Figure 8a, and Model-Tf44, shown in Figure 8b. As shown in the figures, the steel dampers were placed at the perimeter frames to provide torsional resistance. The point G shown in this figure is the center of mass of the superstructure, point S$_0$ is the center of stiffness of the isolation layer calculated according to the initial stiffness of the isolators and dampers, while point S$_1$ is the center of stiffness of the isolation layer calculated according to the secant stiffness of the isolators and dampers considering their displacement ($\delta$) of 0.40 m.

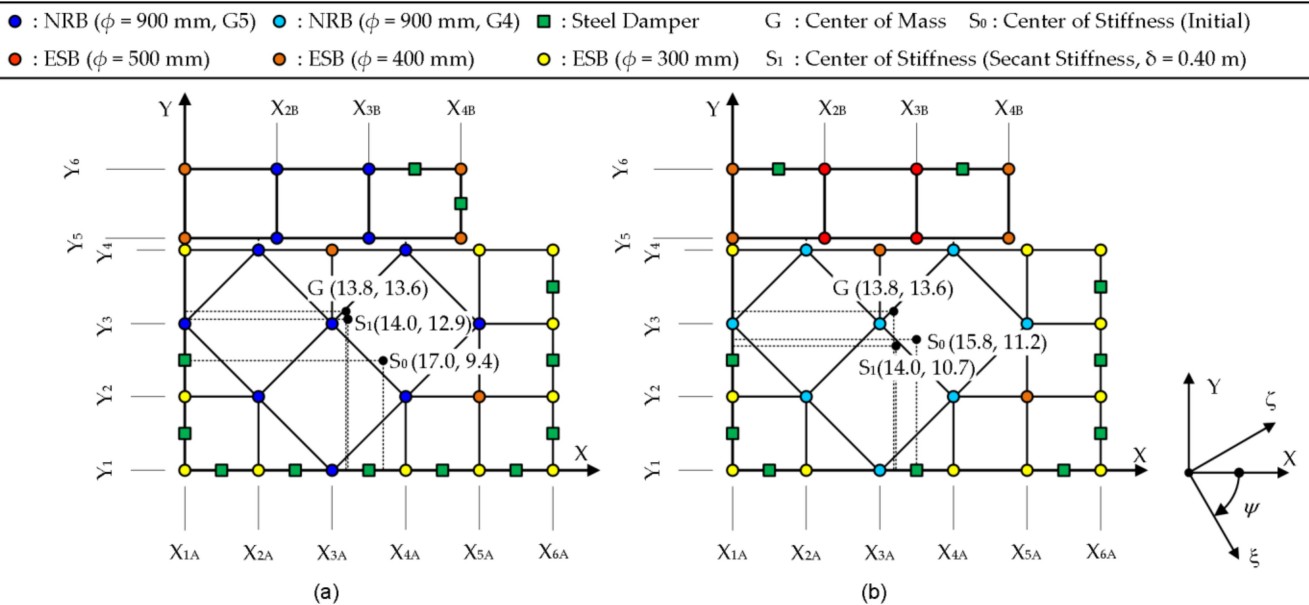

**Figure 8.** Layout of the isolators and dampers in the isolation layer: (**a**) Model-Tf34 and (**b**) Model-Tf44.

For Model-Tf34, $T_f = 3.38$ s and $_s\alpha_y = 0.0448$, while for Model-Tf44, $T_f = 4.40$ s and $_s\alpha_y = 0.0396$. Note that $T_f$ was calculated according to the horizontal stiffness of the NRBs, while $_s\alpha_y$ was calculated according to the yield shear strength of the ESBs and steel dampers.

As shown in Figure 8, the eccentricity at the isolation layer was non-negligible in both models. The eccentricity indices [45] of Model-Tf34, calculated according to the initial stiffnesses in the X and Y directions, were $R_{eX} = 0.285$ and $R_{eY} = 0.216$, respectively, while those calculated according to the secant stiffnesses in the X and Y directions were $R_{eX} = 0.014$ and $R_{eY} = 0.054$, respectively. Similarly, the eccentricity indices of Model-Tf44, calculated according to initial stiffnesses in the X and Y directions, were $R_{eX} = 0.155$ and $R_{eY} = 0.126$, respectively, while those calculated according to the secant stiffnesses in the X and Y directions were $R_{eX} = 0.233$ and $R_{eY} = 0.015$, respectively. In this study, the perimeter frames X$_{1A}$ and Y$_6$ are referred to as the "flexible-side frames," while the frames X$_{6A}$ and Y$_1$ are referred to as the "stiff-side frames". Note that no optimization to

minimize the torsional response was made to choose the dampers in this study, because such optimization was beyond the scope of this study.

Figure 9 shows envelopes of the force–deformation relationship for the isolators and dampers. The behavior of the NRBs was assumed to be linear elastic, while that of the ESBs and dampers was assumed to be bilinear.

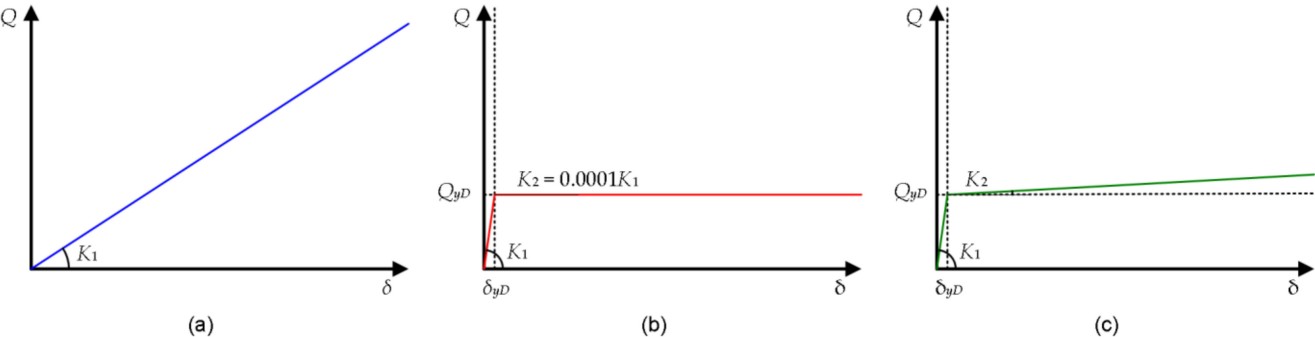

**Figure 9.** Envelope of the force–deformation relationship for the isolators and dampers: (**a**) Natural rubber bearings (NRBs); (**b**) elastic sliding bearings (ESBs); (**c**) steel dampers.

All isolators were chosen from a catalog provided by the Bridgestone Corporation [55,56], considering that the ultimate horizontal deformation was larger than 150% of $\delta_a$ (1.5 × 0.40 m = 0.60 m). Meanwhile, the steel dampers were chosen from a catalog provided by Nippon Steel Corporation Engineering Co. Ltd. (Tokyo, Japan) [57]. Tables 2–4 list the properties of the isolators and dampers used in the two models.

**Table 2.** Properties of the selected natural rubber bearings (NRBs).

| Type | Outer Diameter (mm) | Total Rubber Thickness (mm) | Shear Modulus (MPa) | Horizontal Stiffness $K_1$ (MN/m) | Vertical Stiffness $K_V$ (MN/m) |
|---|---|---|---|---|---|
| NRB ($\phi$ = 900 mm, G5) | 900 | 180 | 0.441 | 1.56 | 3730 |
| NRB ($\phi$ = 900 mm, G4) | 900 | 180 | 0.392 | 1.38 | 3420 |

**Table 3.** Properties of the selected elastic sliding bearings (ESBs).

| Type | Outer Diameter (mm) | Shear Modulus (MPa) | Friction Coefficient $\mu$ | Initial Horizontal Stiffness $K_1$ (MN/m) | Vertical Stiffness $K_V$ (MN/m) |
|---|---|---|---|---|---|
| ESB ($\phi$ = 300 mm) | 300 | 0.392 | 0.010 | 0.884 | 1380 |
| ESB ($\phi$ = 400 mm) | 400 | 0.392 | 0.010 | 1.48 | 2270 |
| ESB ($\phi$ = 500 mm) | 500 | 0.392 | 0.010 | 2.40 | 3710 |

**Table 4.** Properties of the selected steel dampers.

| Initial Stiffness $K_1$ (MN/m) | Yield Strength $Q_{yd}$ (kN) | Post Yield Stiffness $K_2$ (MN/m) |
|---|---|---|
| 7.60 | 184 | 0.128 |

The yield strength of an ESB was calculated as:

$$Q_{yD} = \mu P_V. \tag{15}$$

In Equation (15), $P_V$ is the vertical load of the ESB due to gravity.

### 3.3. Structural Modeling

The two building models were modeled as three-dimensional spatial frames, wherein the floor diaphragms were assumed to be rigid in their own planes without an out-of-plane stiffness. Figure 10 shows the structural modeling. For the numerical analyses, a nonlinear analysis program for spatial frames developed by the authors in a previous study [58] was used. The structural models were based on Model-RuW4-100 from a previous study [44]. As shown in Figure 10b,c, all of the vertical and rotational springs in the basement of the original model were replaced by isolators, and dampers were installed in the isolation layer. The shear behavior of the isolators was modeled using the multi-shear spring model proposed by Wada and Hirose [59], while their axial behavior was modeled using a linear elastic spring, as shown in Figure 10d. The shear behavior of the steel dampers was also modeled using the multi-shear spring model, as shown in Figure 10e. No bending stiffness was considered in either the isolators or the dampers. The hysteresis rules of the ESBs and steel dampers were modeled following the normal bilinear rule for simplicity of analysis. Details of the structural modeling of the superstructure can be found in the literature [44]. The damping matrix of the superstructure was then assumed to be proportional to the tangent stiffness matrix of the superstructure, with 2% of the critical damping of the first mode. The damping of the isolators and dampers was not considered, assuming that their energy absorption effects were already included in the hysteresis rules. Note that the force–displacement relationships of all members, including those in the superstructures and in the isolation layers, were assumed to be symmetric in the positive and negative loading directions, as in previous studies [44,58].

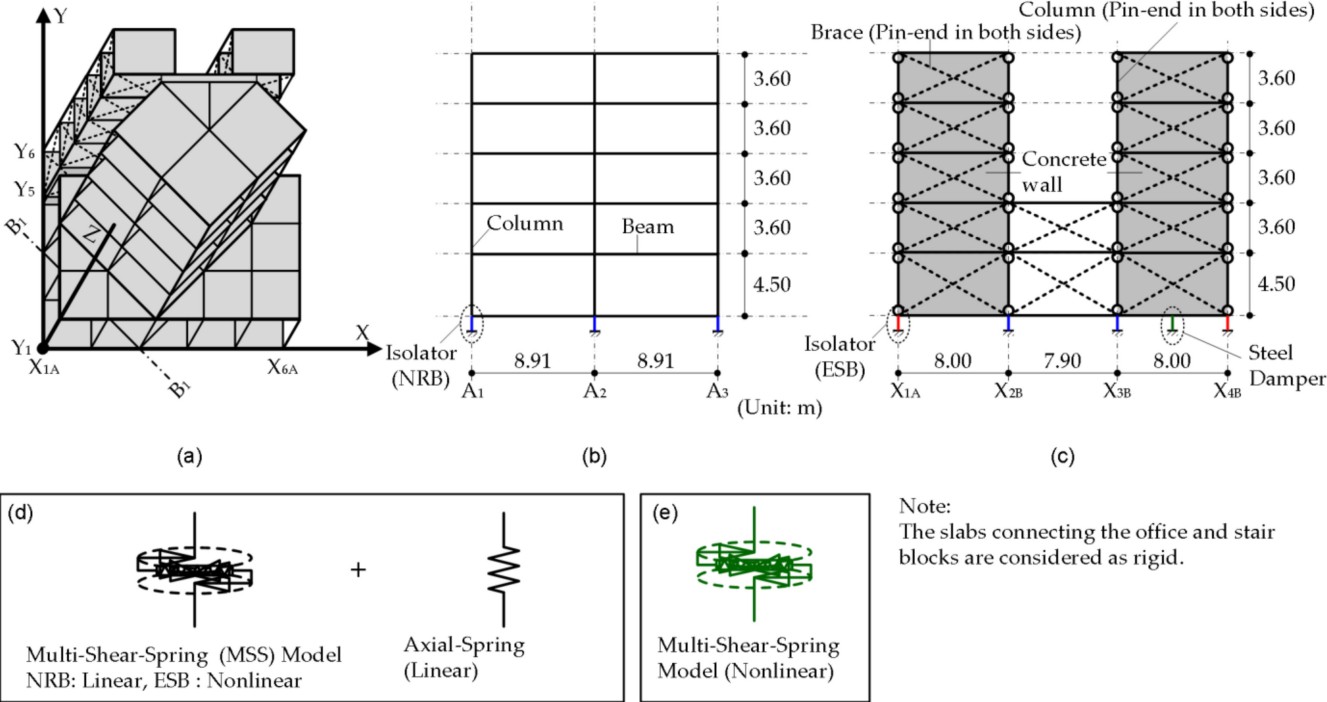

**Figure 10.** Structural modeling: (**a**) Overview of the structural model; (**b**) frame $B_1$; (**c**) frame $Y_6$; (**d**) modeling of the isolators (NRBs and ESBs); (**e**) modeling of the steel dampers.

It should be mentioned that the foundation compliance and kinematic soil–structure interaction were not considered for the simplicity of the analysis. Almansa et al. [60] investigated the performance of a rubber-isolated six-story RC building in soft soil. They concluded that the soil–structure interaction effect is rather negligible in the case of isolated buildings. Based on this, the authors thought the influence of the soil–structure interaction effect to the response of this building would not be significant.

Figures 11 and 12 show the first three natural modes of Model-T34 and Model-T44 in the elastic range. Here, $T_{ie}$ is the natural period of the $i$th mode in the elastic range ($i = 1$–3), $m_{ie}^*$ is the effective modal mass ratio of the $i$th mode with respect to its principal direction in the elastic range, $\psi_{ie}$ is the incidence of the principal direction of the $i$th modal response in the elastic range, and $R_{\rho ie}$ is the torsional index of the $i$th mode in the elastic range. The values of $m_{ie}^*$, $\psi_{ie}$, and $R_{\rho ie}$ were calculated according to the $i$th elastic mode vector ($\boldsymbol{\varphi_{ie}}$) as follows:

$$m_{ie}^* = \frac{1}{M} \frac{\left(\sum\limits_{j=0}^{5} m_j \phi_{Xjie}\right)^2 + \left(\sum\limits_{j=0}^{5} m_j \phi_{Yjie}\right)^2}{\sum\limits_{j=0}^{5} m_j \phi_{Xjie}^2 + \sum\limits_{j=0}^{5} m_j \phi_{Yjie}^2 + \sum\limits_{j=0}^{5} I_j \phi_{\Theta jie}^2}, \tag{16}$$

$$\tan \psi_{ie} = -\sum\limits_{j=0}^{5} m_j \phi_{Yjie} \Big/ \sum\limits_{j=0}^{5} m_j \phi_{Xjie}, \tag{17}$$

$$R_{\rho ie} = \sqrt{\sum\limits_{j=0}^{5} I_j \phi_{\Theta jie}^2 \Big/ \left(\sum\limits_{j=0}^{5} m_j \phi_{Xjie}^2 + \sum\limits_{j=0}^{5} m_j \phi_{Yjie}^2\right)}, \tag{18}$$

$$\boldsymbol{\varphi_{ie}} = \left\{ \phi_{X0ie} \quad \cdots \quad \phi_{X5ie} \quad \phi_{Y0ie} \quad \cdots \quad \phi_{Y5ie} \quad \phi_{\Theta0ie} \quad \cdots \quad \phi_{\Theta5ie} \right\}^{\mathbf{T}}. \tag{19}$$

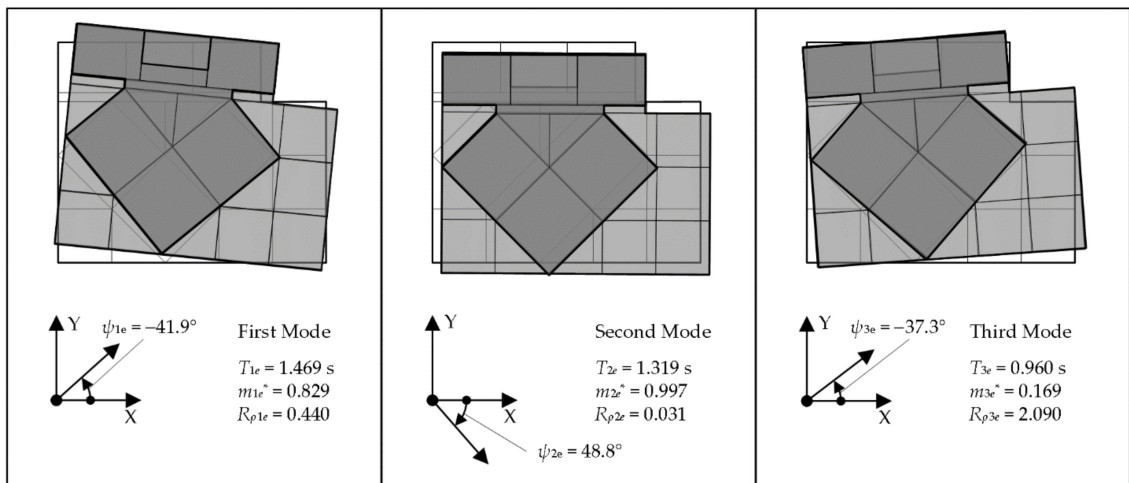

**Figure 11.** Shape of the first three natural modes of Model-Tf34 in the elastic range.

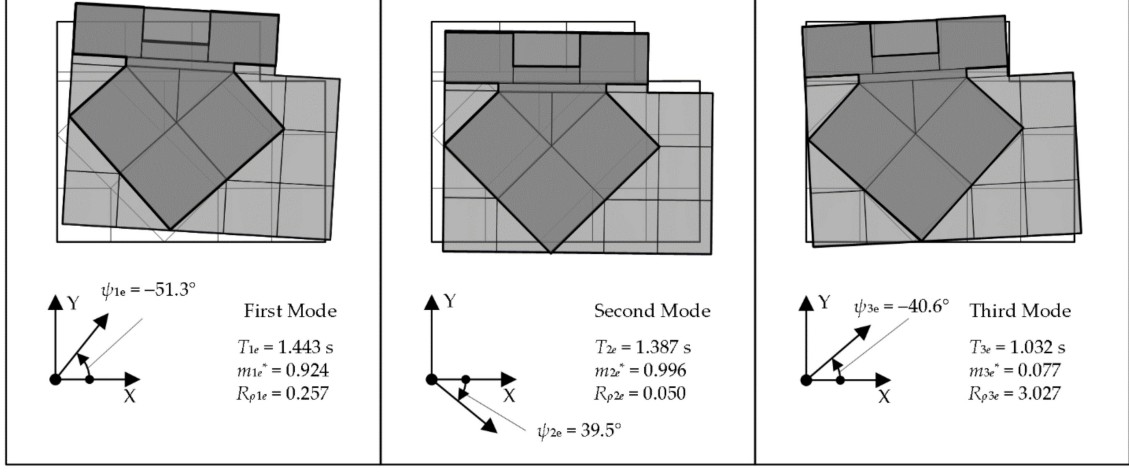

**Figure 12.** Shape of the first three natural modes of Model-Tf44 in the elastic range.

The behavior of the superstructure can be considered a rigid body in the first three modes in the elastic range, as observed from Figures 11 and 12. In both building models, the effective modal mass ratios of the first two modes were close to 1. In addition, the first mode was predominantly translational ($R_{\rho 1e} < 1$), the second mode was nearly purely translational ($R_{\rho 2e} \ll 1$), and the third mode was predominantly torsional ($R_{\rho 3e} > 1$). In addition, the angles between the principal directions of the first two modes were close to 90° (90.7° for Model-Tf34 and 90.8° for Model-Tf44). Therefore, both building models were classified as torsionally stiff buildings: In a previous study [43], the conditions for classification as a torsionally stiff building were $R_{\rho 1}$ and $R_{\rho 2} < 1$, and $R_{\rho 3} > 1$.

The natural periods of the superstructure in the elastic range, assuming that the entire basement is supported by pins, were 0.373 s (first mode), 0.342 s (second mode), and 0.192 s (third mode). Therefore, the natural period of the superstructure was well separated from the seismic isolation period in both models.

### 3.4. Ground Motion Data

In this study, the seismic excitation was bidirectional in the X–Y plane, and both artificial and recorded ground motions were considered.

### 3.4.1. Artificial Ground Motions

In this study, two series of artificial ground motion datasets (the Art-1 and Art-2 series) were generated. As noted in Section 3.2, the equivalent velocity of the total input energy ($V_I$) was set to 1.41 m/s to determine the properties of the isolation layer. Therefore, the artificial ground motion datasets were generated such that their bidirectional $V_I$ spectrum fit the predetermined target $V_I$ spectrum as follows.

The target bidirectional $V_I$ spectrum with a damping ratio of 0.10 was determined such that:

$$V_I(T) = \begin{cases} 1.41(T/0.75) \text{ m/s} & : T \leq 0.75 \text{ s} \\ 1.41 & : T > 0.75 \text{ s} \end{cases} . \tag{20}$$

To generate each horizontal component, it was assumed that the intensity of $V_I$ in both orthogonal components ($\xi$ and $\zeta$ components) was identical. The target equivalent velocities of the total input energy of the $\xi$ and $\zeta$ components, $V_{I\xi}(T)$ and $V_{I\zeta}(T)$, respectively, were calculated as:

$$V_{I\xi}(T) = V_{I\zeta}(T) = V_I(T)/\sqrt{2}. \tag{21}$$

The phase angle is given by a uniform random value, and to consider the time-dependent amplitude of the ground motions, a Jenning-type envelope function ($e(t)$) was assumed. In this study, two envelope functions were considered. In Art-1-00, the envelope function was set as in Equation (22), while in Art-2-00, the envelope function was set as in Equation (23).

$$e(t) = \begin{cases} (t/5)^2 & : 0 \text{ } s \leq t \leq 5 \text{ } s \\ 1 & : 5 \text{ } s \leq t \leq 25 \text{ } s \\ \exp\{-0.066(t-25)\} & : 25 \text{ } s \leq t \leq 60 \text{ } s \end{cases} \tag{22}$$

$$e(t) = \begin{cases} (t/2.5)^2 & : 0 \text{ } s \leq t \leq 2.5 \text{ } s \\ 1 & : 2.5 \text{ } s \leq t \leq 12.5 \text{ } s \\ \exp\{-0.132(t-12.5)\} & : 12.5 \text{ } s \leq t \leq 30 \text{ } s \end{cases} \tag{23}$$

Figures 13 and 14 show the time-histories of the two components and the orbits of Art-1-00 and Art-2-00, respectively. Because the same target $V_I$ spectrum was applied while the duration of Art-2-00 is the half that of Art-1-00, the peak ground acceleration of Art-2-00 was larger than that of Art-1-00.

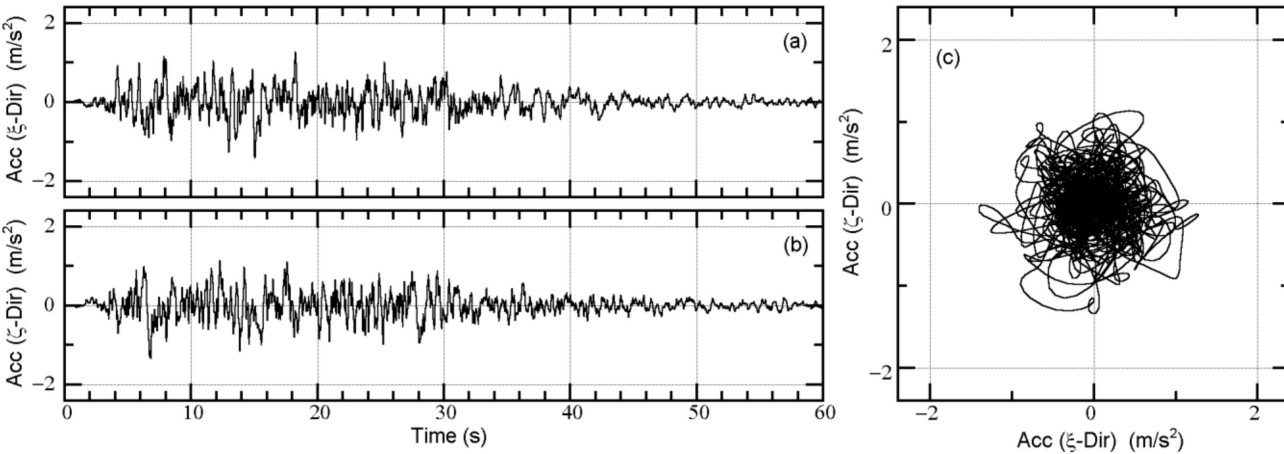

**Figure 13.** Two components of the generated artificial ground motion (Art-1-00): (**a**) $\xi$ direction; (**b**) $\zeta$ direction; (**c**) orbit.

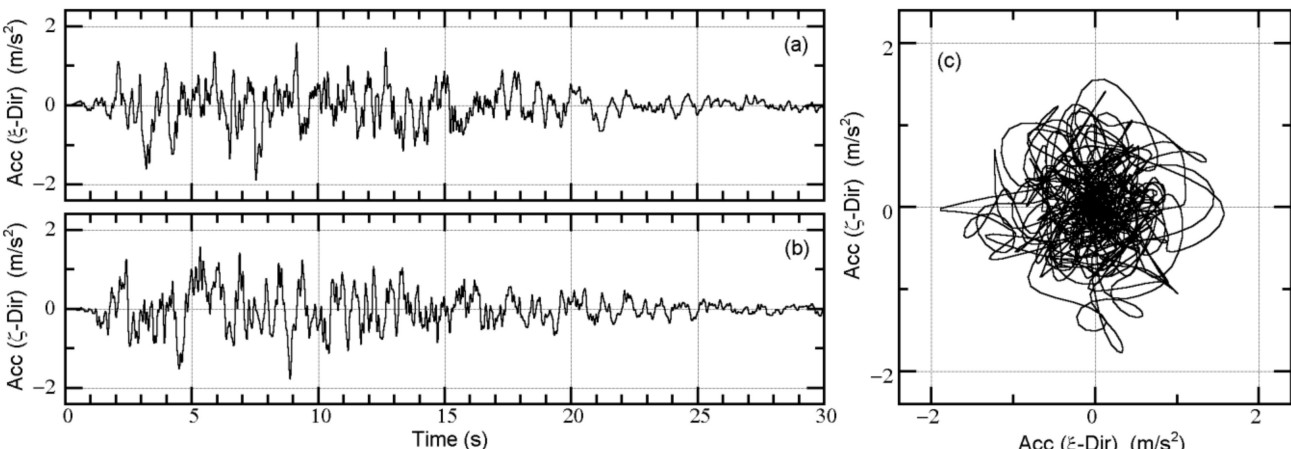

**Figure 14.** Two components of the generated artificial ground motion (Art-2-00): (**a**) $\xi$ direction; (**b**) $\zeta$ direction; (**c**) orbit.

Note that the artificial ground motions used in this study were generated independently. The correlation coefficients of Art-1-00 and Art-2-00 were close to zero, even though the envelope functions of the two components were the same; therefore, the two components can be considered independently of one another.

Next, 11 artificial ground motion datasets were generated from Art-1-00 and Art-2-00 by shifting the phasing angle. The generated artificial ground motion vector ($\mathbf{a_g}(t, \Delta\phi_0)$) is expressed as:

$$\mathbf{a_g}(t, \Delta\phi_0) = \left\{ \begin{array}{c} a_{g\xi}(t, \Delta\phi_0) \\ a_{g\zeta}(t, \Delta\phi_0) \end{array} \right\} = \sum_{n=-N}^{N} \left\{ \begin{array}{c} c_{\xi,n} \\ c_{\zeta,n} \end{array} \right\} \exp[i\{\omega_n t - \mathrm{sgn}(\omega_n)\Delta\phi_0\}]. \tag{24}$$

In Equation (24), $c_{\xi,n}$ and $c_{\zeta,n}$ are the complex Fourier coefficients of the *n*th harmonics of $a_{g\xi}(t)$ and $a_{g\zeta}(t)$, respectively, $\omega_n$ is the circular frequency of the *n*th harmonic, and $\Delta\phi_0$ is the constant used to shift the phase angle of all the harmonics. As in previous studies [51,53], the constant $\Delta\phi_0$ was set from $\pi/12$ to $11\pi/12$ with an interval of $\pi/12$. Notably, the phase difference of each ground motion component did not change when shifting the phase angle. In addition, the total input energy spectrum (the $V_I$ spectrum) was independent of the shifting phase angle ($\Delta\phi_0$). This is because the shifting phase angle does not affect the absolute value of the complex Fourier coefficient, and the total input energy per unit mass can be calculated from the Fourier amplitude spectrum without the phase characteristics of the ground motion, as shown by Ordaz [61]. The generated

artificial ground motion datasets are numbered from 01 to 11 depending on $\Delta\phi_0$, with a total of $2 \times 12 = 24$ artificial ground motion datasets generated and used in this study.

Figures 15 and 16 show the horizontal response spectra of the generated ground motion datasets. As shown in these figures, the spectral acceleration ($S_A$) and the spectral velocity ($S_V$) differed slightly because of the shift in the phase angle. A comparison of Figures 15 and 16 reveals that $S_A$ and $S_V$ of the Art-2 series were larger than those of the Art-1 series, even though the target $V_I$ spectrum was identical.

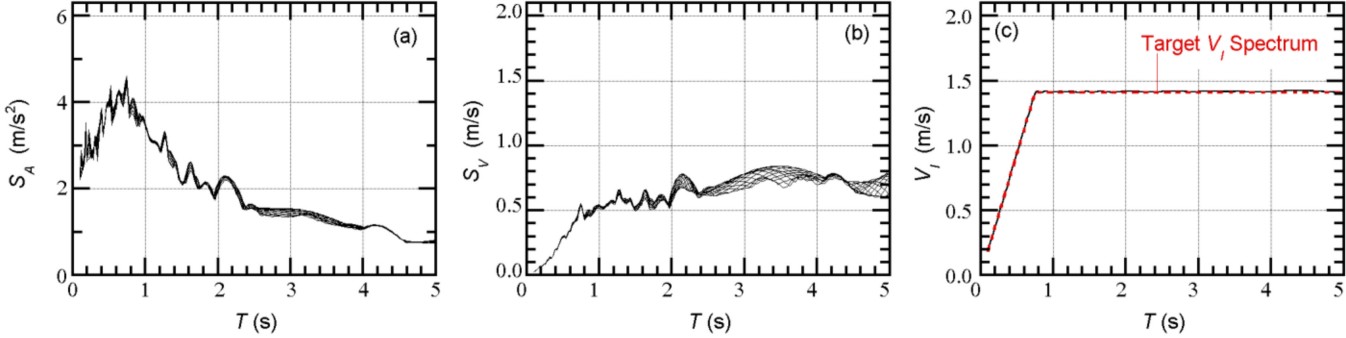

**Figure 15.** Horizontal response spectra of the artificial ground motion (Art-1 series): (**a**) Elastic acceleration response spectrum (damping ratio: 0.05); (**b**) elastic velocity response spectrum (damping ratio: 0.05); (**c**) elastic total input energy spectrum (damping ratio: 0.10).

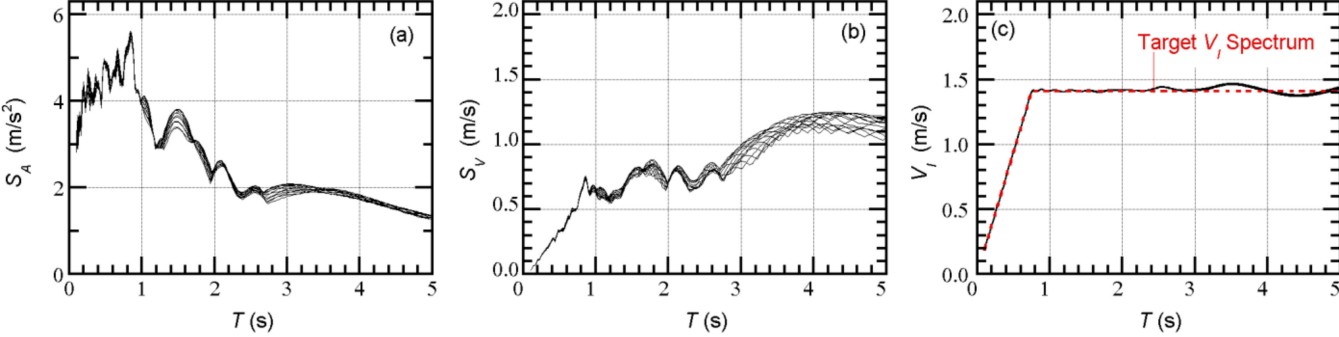

**Figure 16.** Horizontal response spectra of the artificial ground motion (Art-2 series): (**a**) Elastic acceleration response spectrum (damping ratio: 0.05); (**b**) elastic velocity response spectrum (damping ratio: 0.05); (**c**) elastic total input energy spectrum (damping ratio: 0.10).

### 3.4.2. Recorded Ground Motions

Table 5 lists the four datasets of recorded ground motions used in this study. Details concerning the original ground motions can be found in the Appendix A. Figure 17 shows the horizontal response spectra of the unscaled recorded ground motion datasets.

**Table 5.** List of the recorded ground motion datasets used in this study.

| Earthquake of the Original Record | Ground Motion ID | Scale Factor | |
|---|---|---|---|
| | | Model-Tf34 | Model-Tf44 |
| Kumamoto, 14 April 2016 | UTO0414 | 1.000 | 1.000 |
| Kumamoto, 16 April 2016 | UTO0416 | 1.000 | 1.000 |
| Chichi, 1999 | TCU | 0.5540 | 0.5718 |
| Kocaeli, 1999 | YPT | 0.4293 | 0.5057 |

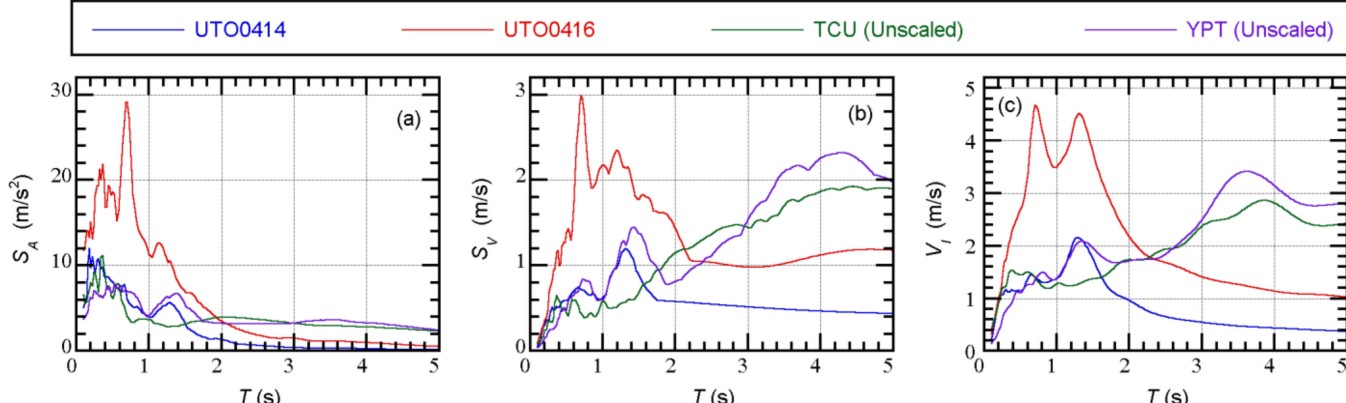

**Figure 17.** Horizontal response spectra of the recorded ground motions: (**a**) Elastic acceleration response spectrum (damping ratio: 0.05); (**b**) elastic velocity response spectrum (damping ratio: 0.05); (**c**) elastic total input energy spectrum (damping ratio: 0.10).

One of the motivations of this analysis was to evaluate the seismic performance of the two retrofitted models under the ground motions recorded during the 2016 Kumamoto earthquake. Therefore, the two recorded ground motion datasets (UTO0414 and UTO0416) [54] were used without scaling. A further two datasets of ground motions, TCU and YPT, were chosen as examples of long-period pulse-like ground motions, as investigated by Güneş and Ulucan [46]. These two ground motion datasets were scaled such that the equivalent velocity of the total energy input ($V_I$) at the isolation period ($T_f$) equaled 1.41 m/s, which is consistent with the intensity considered in the determination of the properties of the isolation layer, as discussed in Section 3.2.

## 4. Analysis Results

The analysis results are split into two subsections. In the first subsection, an example of the prediction of the peak equivalent displacement using the updated MABPA is shown. In the second subsection, comparisons are made between the nonlinear time-history analysis results and the predicted results.

### 4.1. Example of a Prediction of the Peak Equivalent Displacement

The structural model shown here as an example is Model-Tf44. Figure 18 shows the nonlinear properties of the equivalent SDOF model calculated according to Section 2.2.2.

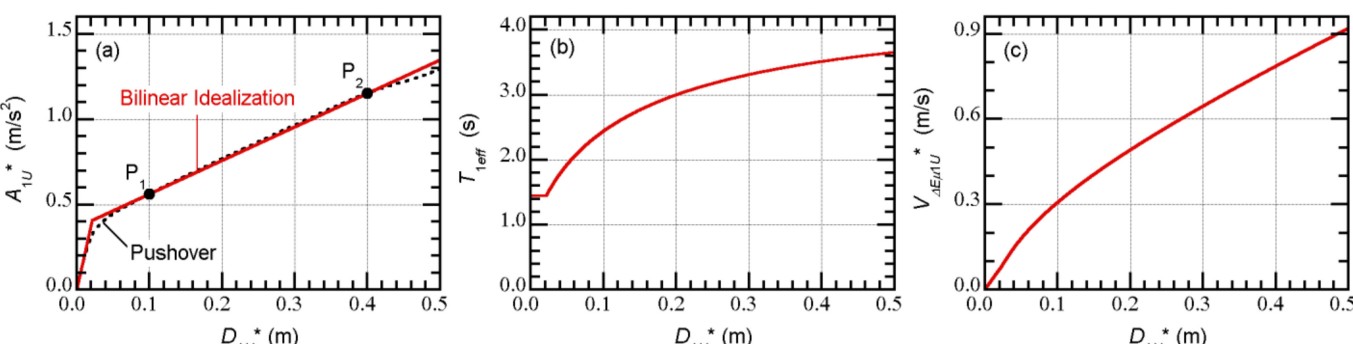

**Figure 18.** Nonlinear properties of the equivalent single-degree-of-freedom (SDOF) model representing the first modal response (Model-Tf44): (**a**) The $A_{1U}{}^*-D_{1U}{}^*$ relationship; (**b**) the $T_{1eff}{}^*-D_{1U}{}^*$ relationship; (**c**) the $V_{\Delta E\mu 1U}{}^*-D_{1U}{}^*$ relationship.

In the results shown in Figure 18b,c, the $V_{\Delta E\mu 1U}{}^*-T_{1eff}$ curve was constructed and overlaid with the bidirectional $V_{\Delta E}$ spectrum. Figure 19 shows the prediction of the peak equivalent displacements of the first two modes.

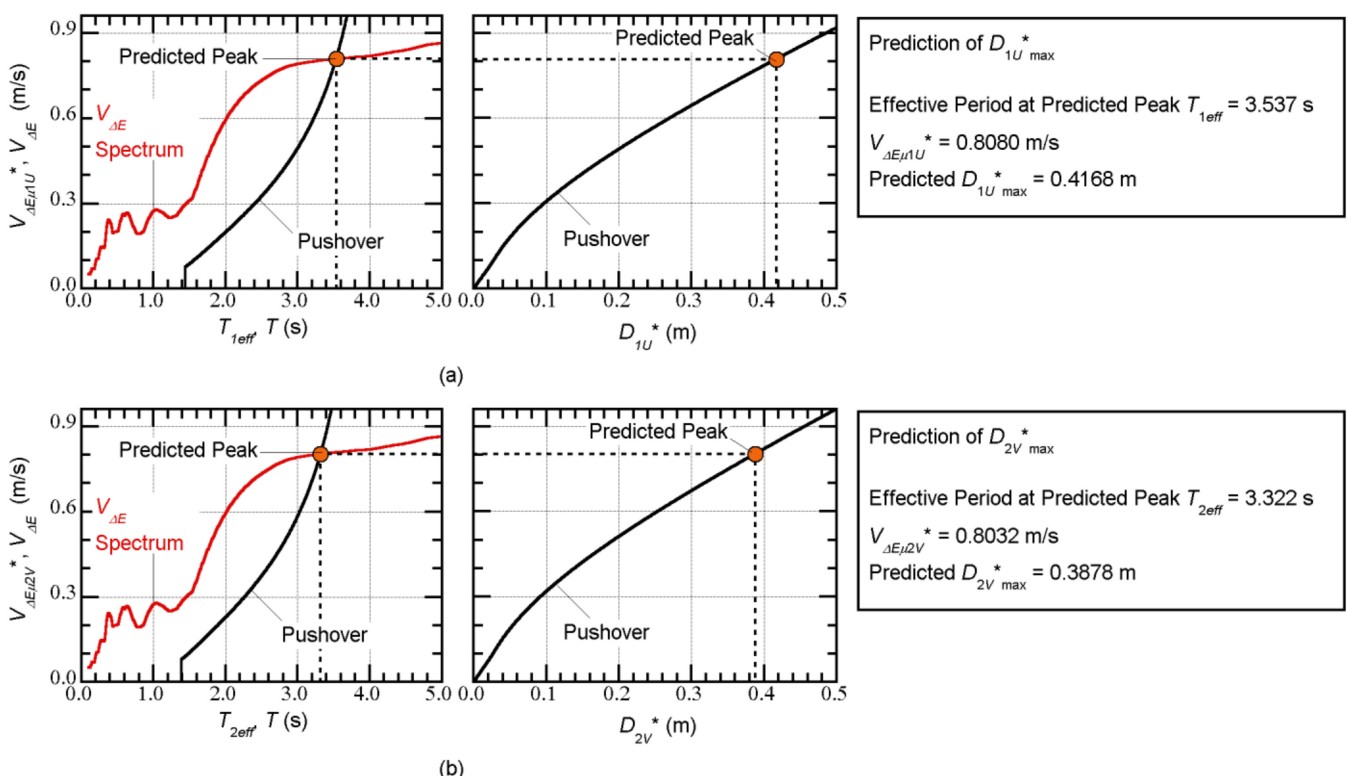

**Figure 19.** Prediction of the peak equivalent displacements (model: Model-Tf44, ground motion: TCU): (**a**) Prediction of the peak of the first modal response and (**b**) prediction of the peak of the second modal response.

The predicted peak was obtained as the intersection point of the $V_{\Delta E\mu 1U}{}^{*}-T_{1eff}$ curve according to the pushover analysis results and the bidirectional $V_{\Delta E}$ spectrum, as shown on the left side of Figure 19a. Then, the peak equivalent displacement $D_{1U}{}^{*}{}_{max}$ was predicted from the $V_{\Delta E\mu 1U}{}^{*}-D_{1U}{}^{*}{}_{max}$ curve. The prediction of the peak equivalent displacement of the second mode, $D_{2V}{}^{*}{}_{max}$, can be made using the same procedure as that of $D_{1U}{}^{*}{}_{max}$, as shown in Figure 19b. Note that for the predictions of $D_{1U}{}^{*}{}_{max}$ and $D_{2V}{}^{*}{}_{max}$, the same $V_{\Delta E}$ spectrum (bidirectional $V_{\Delta E}$ spectrum) was used.

*4.2. Comparisons with the Nonlinear Time-History Analysis*

In Section 4.2.1, the nonlinear time-history analysis results using 24 artificial ground motion datasets are compared with the predicted results. In this analysis, 24 × 4 = 96 cases were analyzed for each model: The angle of incidence of the horizontal ground motion ($\psi$) in Figure 3 was set to −45°, 0°, 45°, and 90° considering the symmetry of the force-deformation relationship assumed in the structural model. Because the characteristics of the two horizontal components were similar, the discussion focuses on comparisons between the envelope of the time-history analysis results and the predicted results.

In Section 4.2.2, the nonlinear time-history analysis results using the four recorded ground motion datasets are compared with the predicted results. In this analysis, 4 × 12 = 48 cases were analyzed for each model: The angle of incidence of the horizontal ground motion ($\psi$) was set to values from −75° to 90° with an interval of 15°.

4.2.1. Artificial Ground Motion

Figures 20 and 21 show the peak relative displacement at $X_{3A}Y_3$, which is the closest point to the center of the floor mass at each level. In these figures, the nonlinear time-history analysis results are compared with the results predicted by MABPA (for complex damping ratios ($\beta$) of 0.10, 0.20, and 0.30). As shown here, the predicted peaks can approximate the envelope of the time-history analysis results, except in the case of Model-Tf34 subjected to

the Art-2 series shown in Figure 20b. In addition, the predicted peak with $\beta = 0.10$ is larger than that with $\beta = 0.30$. When the models were subjected to the Art-1 series, the predicted peak closest to the envelope of the time-history analysis results was found with $\beta = 0.10$ for Model-Tf34 (Figure 20a) and with $\beta = 0.30$ for Model-Tf44 (Figure 21b). Differences in the predicted peaks resulting from the value of the complex damping were small in the case of Model-Tf34 but relatively noticeable in the case of Model-Tf44.

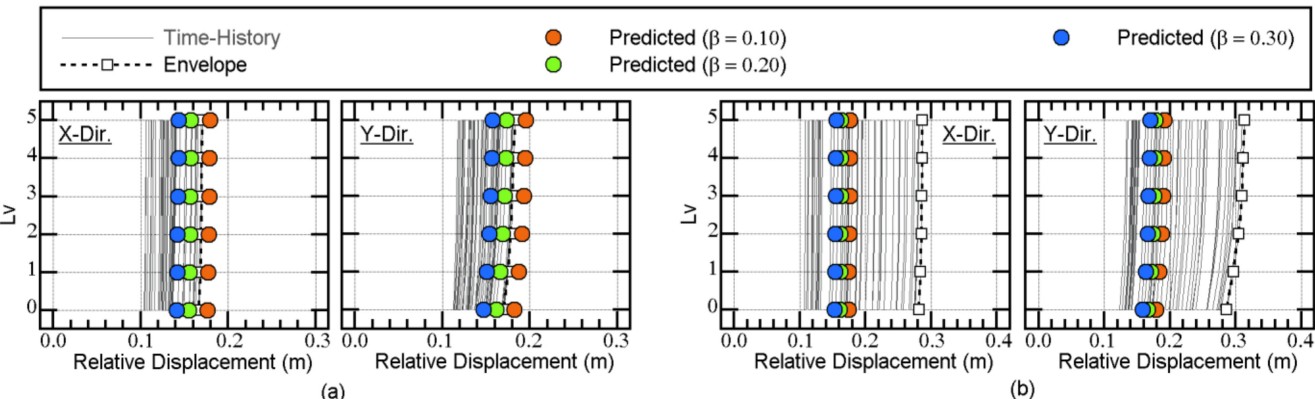

**Figure 20.** Comparison of the peak relative displacements at $X_{3A}Y_3$ (Model-Tf34) for the (**a**) Art-1 and (**b**) Art-2 series.

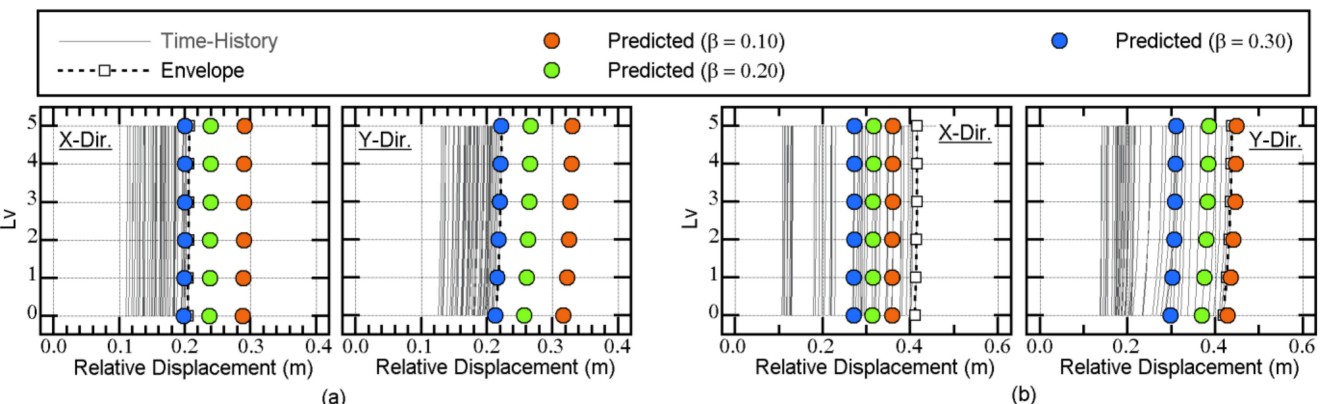

**Figure 21.** Comparison of the peak relative displacements at $X_{3A}Y_3$ (Model-Tf44) for the (**a**) Art-1 and (**b**) Art-2 series.

Figures 22 and 23 show comparisons of the horizontal distributions of the peak displacement at level 0, which is just above the isolation layer. In Figure 22, the results of Model-Tf34 subjected to the Art-1 series are shown. The predicted peak agrees with the envelope of the time-history analysis results, even though the predicted peak was not conservative at $Y_1$ and $Y_2$ (the stiff side in the X direction). A similar observation can be made in the case of Model-Tf44 subjected to the Art-1 series, as shown in Figure 23, where the predicted peak with $\beta = 0.30$ agrees very well with the envelope of the time-history analysis results, except at $Y_1$ and $Y_2$.

Based on the above results, the updated MABPA can predict the peak relative displacement at $X_{3A}Y_3$ and the horizontal distribution of the peak displacement at level 0 for both models according to the artificial ground motion datasets.

### 4.2.2. Recorded Ground Motion

Figures 24 and 25 show the peak relative displacement at $X_{3A}Y_3$. The accuracy of the predicted peak displacement is satisfactory: The predicted peak approximated the envelope of the nonlinear time-history analysis results, even though some cases were overestimated (e.g., model: Model-Tf44, ground motion: UTO0416, Figure 25b). The variation of the predicted peak due to the assumed complex damping ratio ($\beta$) depends on the ground

motion. In some cases, the largest peak was obtained when *β* was 0.30 (e.g., model: Model-Tf34, ground motion: UTO0416, Figure 24b), while in other cases, the largest peak was obtained when *β* was 0.10 (e.g., model: Model-Tf34, ground motion: TCU, Figure 24c).

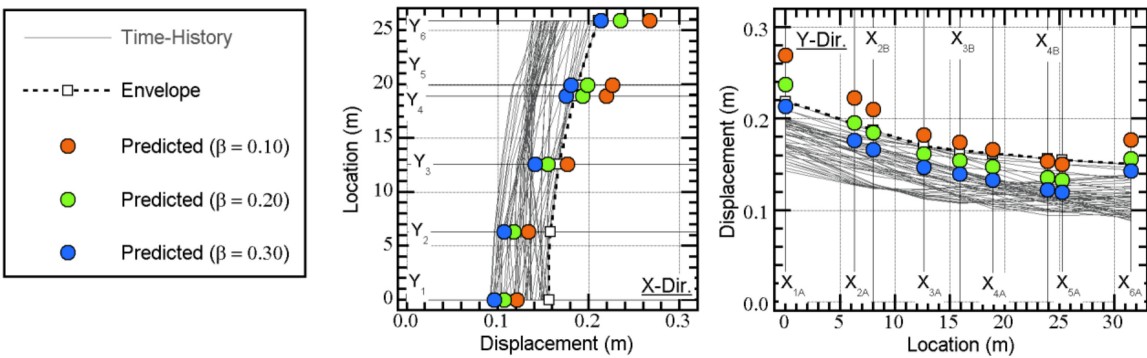

**Figure 22.** Comparison of the horizontal distributions of the peak displacement at level 0 (model: Model-Tf34, ground motion: Art-1 series).

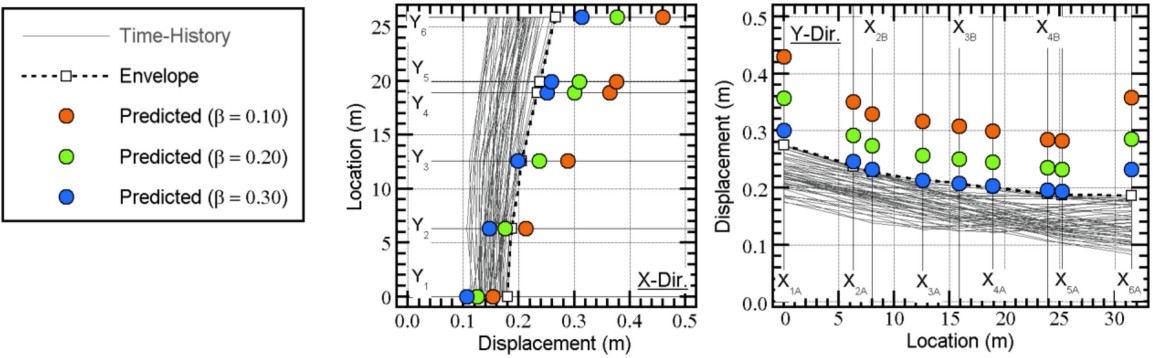

**Figure 23.** Comparison of the horizontal distributions of the peak displacement at level 0 (model: Model-Tf44, ground motion: Art-1 series).

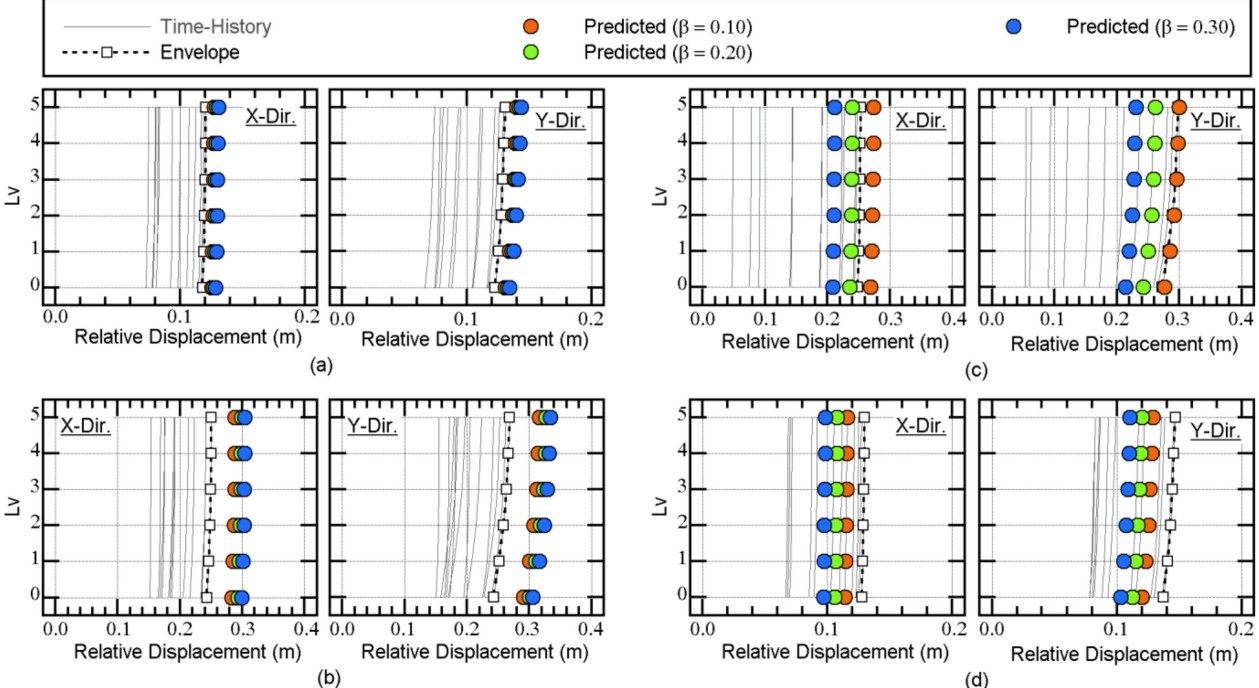

**Figure 24.** Comparison of the peak relative displacements at $X_{3A}Y_3$ (Model-Tf34): (**a**) UTO0414; (**b**) UTO0416; (**c**) TCU; (**d**) YPT.

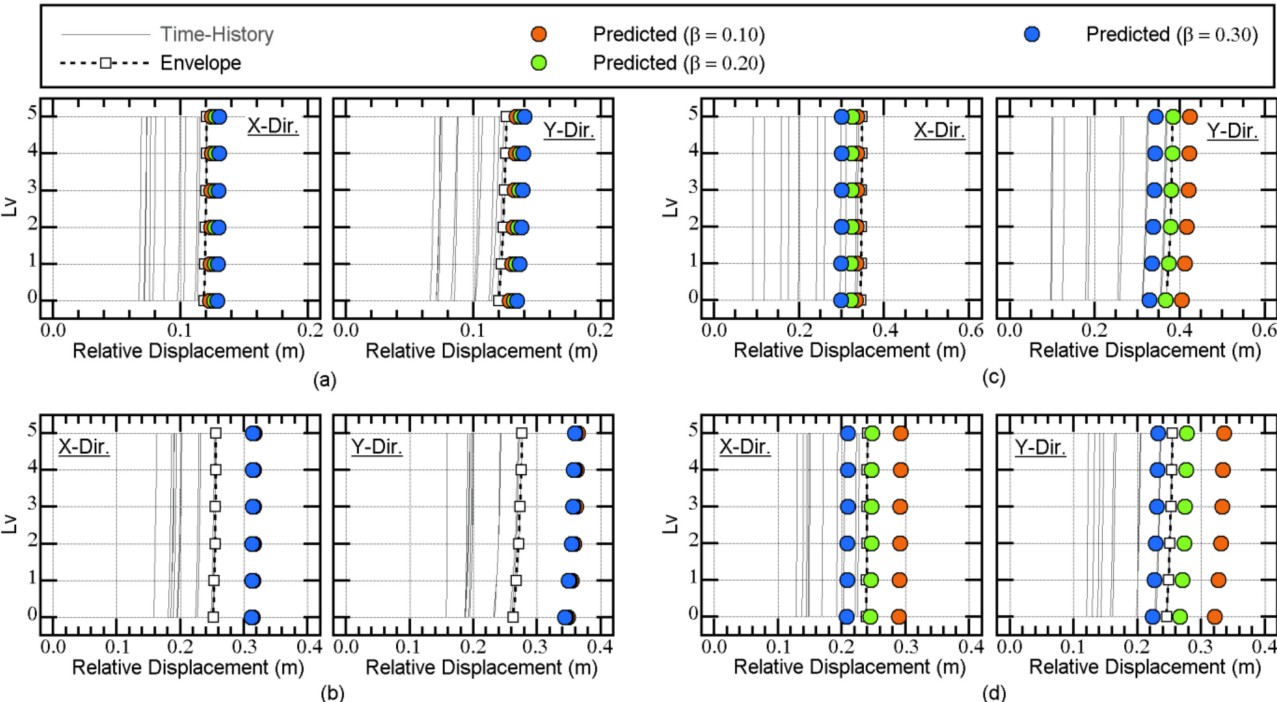

**Figure 25.** Comparison of the peak relative displacements at $X_{3A}Y_3$ (Model-Tf44): (**a**) UTO0414; (**b**) UTO0416; (**c**) TCU; (**d**) YPT.

Figure 26 shows comparisons of the horizontal distribution of the peak displacement at level 0. In this figure, the Model-Tf34 results are shown, with the results for UTO0414 shown in Figure 26a and those for TCU shown in Figure 26b.

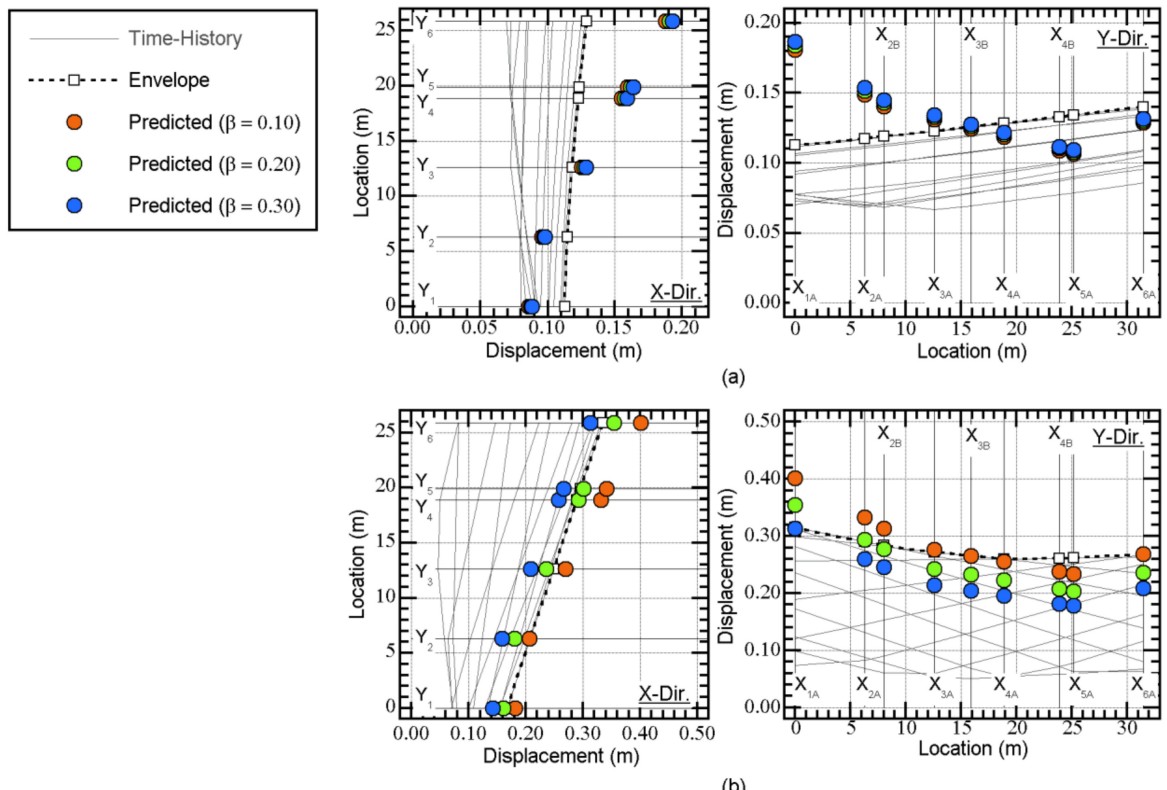

**Figure 26.** Comparison of the horizontal distributions of the peak displacement at level 0 (model: Model-Tf34) for (**a**) UTO0414 and (**b**) TCU.

The predicted horizontal distribution of the peak displacement in the Y direction was notably different from the envelope of the time-history analysis in the case of UTO0414, as shown in Figure 26a. In the predicted distribution, the largest displacement occurred at $X_{1A}$ (the flexible side in the Y direction); meanwhile, in the envelope of the time-history analysis results, the displacement at $X_{1A}$ was the smallest. Conversely, the predicted horizontal distribution of the peak displacement in the Y direction fits the envelope of the time-history analysis in the case of TCU very well, as shown in Figure 26b. In the predicted distribution, the largest displacement occurred at $X_{1A}$, which is consistent with the envelope of the time-history analysis results. The modal displacement responses at $X_{1A}$ and $X_{6A}$ (the stiff side in the Y direction) are compared and discussed further in Section 5.5.

One of the motivations of this analysis was to evaluate the seismic performance of the retrofitted building models under the recorded motions from the 2016 Kumamoto earthquake. It has already been reported in a previous study [44] that the seismic capacity evaluation of the main building of the former Uto City Hall indicate that the structure was insufficient to withstand the earthquake that occurred on 14 April 2016. Therefore, the drift of the superstructure was examined as follows.

Figure 27 shows comparisons of the peak drift for the three columns in Model-Tf34 in cases with UTO0414 and UTO0416. The peak drifts for the three columns obtained from the nonlinear time-history analysis results were smaller than 0.4%. The largest peak drift occurred for column $A_3B_3$ on the second story, with a value of 0.12% in the case of UTO0414 (Figure 27a) and a value of 0.30% in the case of UTO0416 (Figure 27b). Therefore, the seismic performance of Model-Tf34 was excellent for the motions recorded during the 2016 Kumamoto earthquake. Figure 28 shows comparisons of the peak drift of Model-Tf44. Similar observations can be made for Model-Tf44. Comparisons of Figures 27 and 28 indicate that the peak drift of Model-Tf44 was smaller than that of Model-Tf34. These figures also illustrate that the accuracy of the predicted peak depends on the column, with the accuracy in column $A_1B_1$ being satisfactory and that in column $A_3B_3$ being less satisfactory.

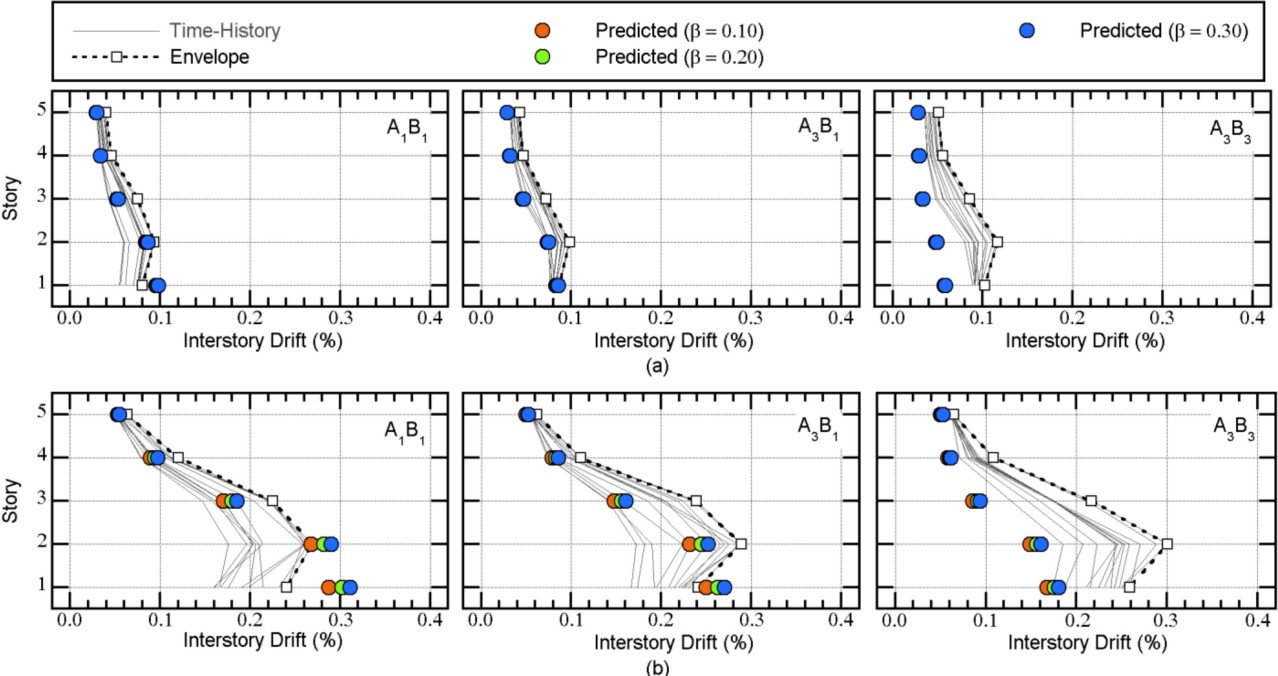

**Figure 27.** Comparison of the peak drift for the columns (Model-Tf34) with (**a**) UTO0414 and (**b**) UTO0416.

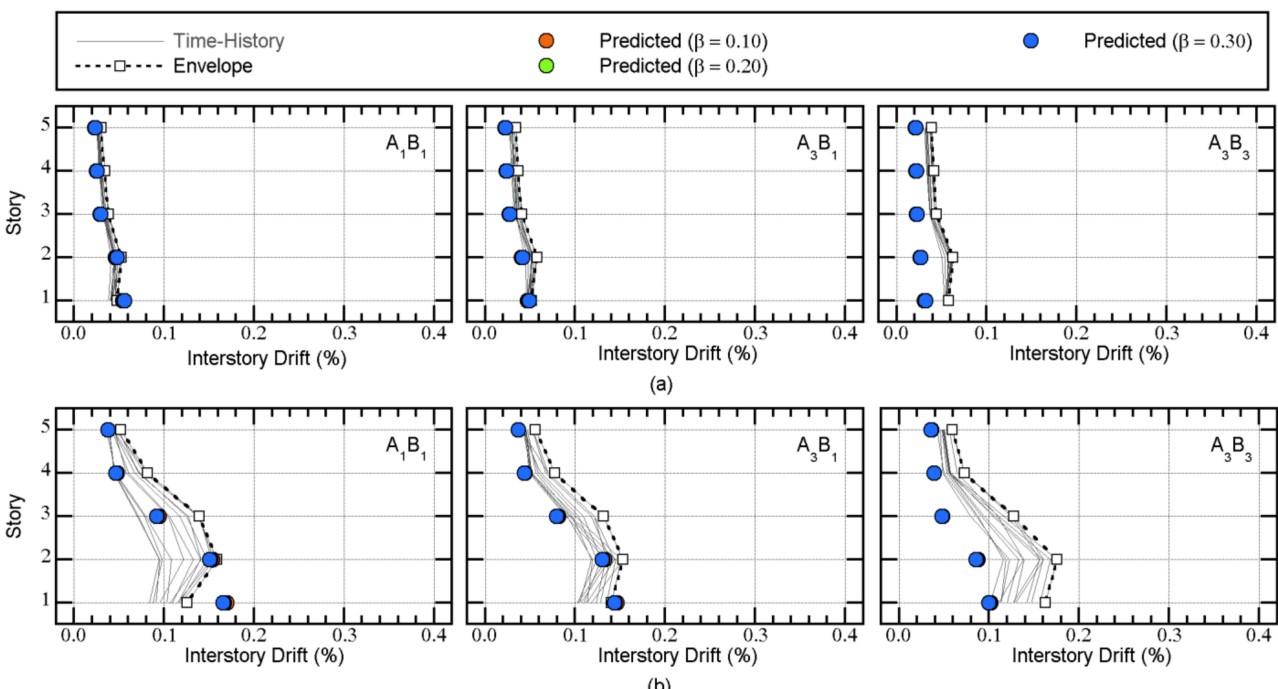

**Figure 28.** Comparison of the peak drift for the columns (Model-Tf44) with (**a**) UTO0414 and (**b**) UTO0416.

Figures 29 and 30 show the peak displacements at two isolators ($X_{1A}Y_6$ and $X_{6A}Y_1$) for various angles of incidence of seismic input ($\psi$). In these figures, the displacement of each isolator was calculated as the absolute (vector) value of the two horizontal directions, and the predicted peaks are shown by the colored lines. As shown in these figures, the upper bounds of the peak displacement of the isolators can be satisfactorily evaluated using the updated MABPA presented in this study.

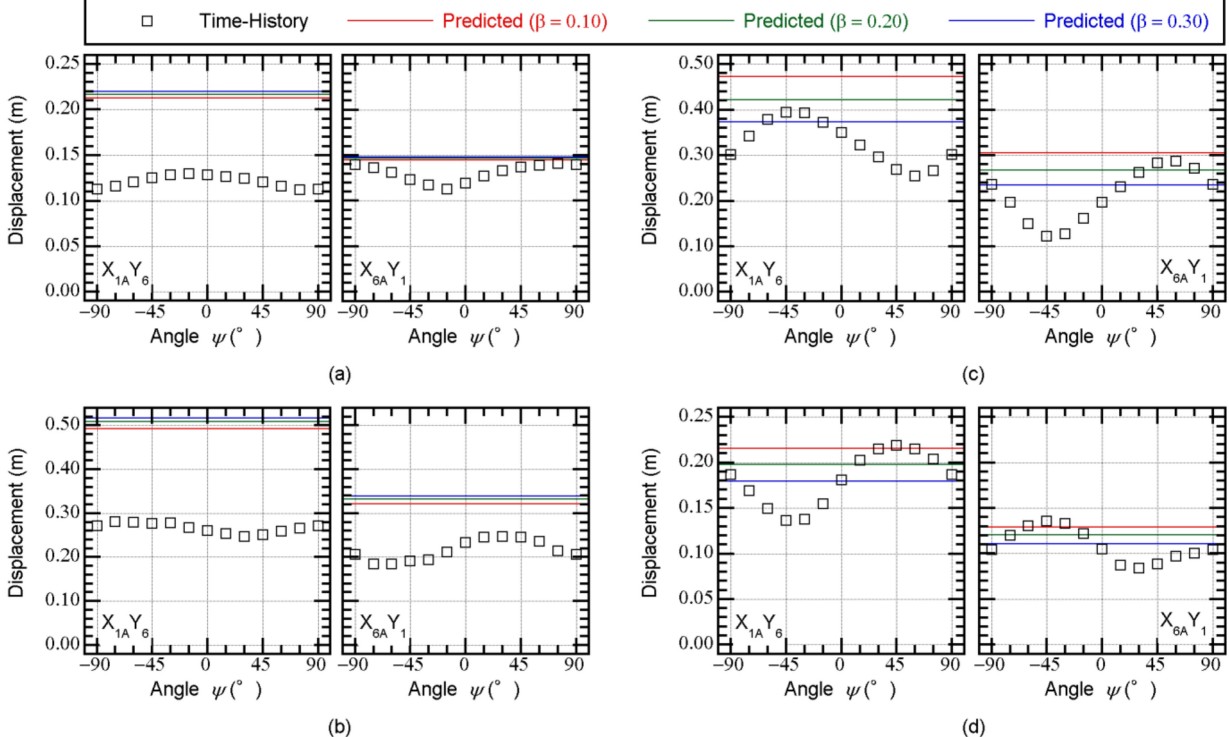

**Figure 29.** Peak displacement at isolators for various angles of incidence of seismic input (Model-Tf34): (**a**) UTO0414; (**b**) UTO0416; (**c**) TCU; (**d**) YPT.

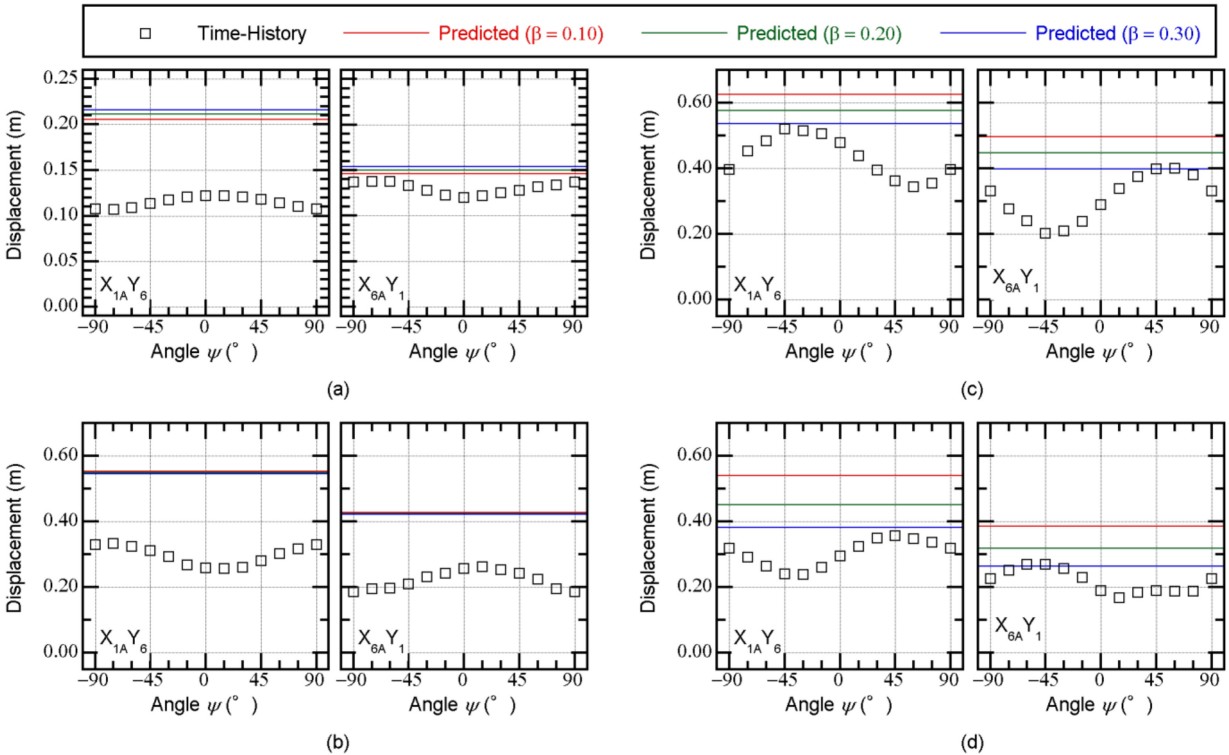

**Figure 30.** Peak displacement at isolators for various angles of incidence of seismic input (Model-Tf44): (**a**) UTO0414; (**b**) UTO0416; (**c**) TCU; (**d**) YPT.

These figures also indicate that the variation in the peak displacement of the isolators due to the angle of incidence of the seismic input is noticeable, e.g., in the cases of TCU and YPT, for both models. This point is discussed in Section 5.

## 5. Discussion

Here, there are four points of focus: (i) The relationship between the maximum momentary input energy and the peak equivalent displacement of the first modal response and its predictability from the pushover analysis results, (ii) the predictability of the maximum momentary input energy of the first modal response from the bidirectional $V_{\Delta E}$ spectrum, (iii) the accuracy of the predicted peak equivalent displacements of the first and second modal responses, and (iv) the contribution of the higher mode to the displacement response at the edge of level 0.

### 5.1. Calculation of the Modal Responses

Figure 31 shows the calculation flow for the first and second modal responses from the nonlinear time-history analysis results. This procedure is based on the procedure originally proposed by Kuramoto [62] for analyzing the nonlinear modal response of a non-isolated planer frame structure extended to analyze that of three-dimensional base-isolated structures.

Next, the momentary input energy of the first modal response per unit mass was calculated as follows. From the time-history of the equivalent displacement of the first modal response ($D_{1U}{}^*(t)$), the momentary input energy of the first modal response per unit mass was calculated as:

$$\frac{\Delta E_{1U}{}^*}{M_{1U}{}^*} = -\int_{t}^{t+\Delta t} \dot{D}_{1U}{}^*(t) a_{gU}(t) dt. \tag{25}$$

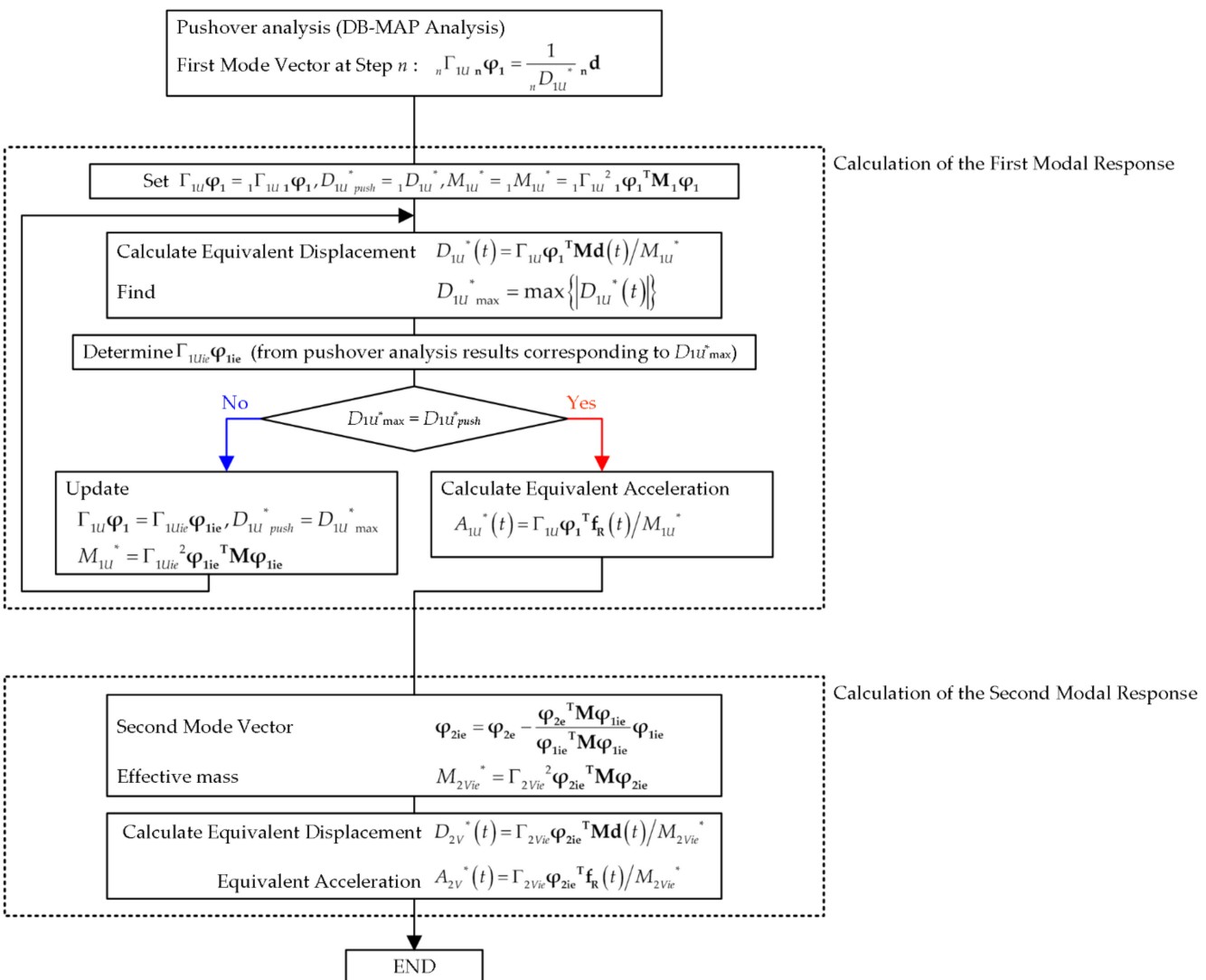

**Figure 31.** Calculation flow for the first and second modal responses.

In Equation (25), $\dot{D}_{1U}{}^*(t)$ is the derivative of $D_{1U}{}^*(t)$ with respect to $t$, $a_{gU}(t)$ is the ground acceleration component of the U-axis (the principal axis of the first modal response corresponds to $D_{1U}{}^*{}_{\max} = \max\{|D_{1U}{}^*(t)|\}$), and $\Delta t$ is the duration of a half cycle of the first modal response. The maximum momentary input energy of the first modal response per unit mass ($\Delta E_{1U}{}^*{}_{\max}/M_{1U}{}^*$) is the maximum value of $\Delta E_{1U}{}^*/M_{1U}{}^*$ calculated by Equation (25) over the course of the seismic event. Figure 32 shows the definition of the maximum momentary input energy of the first modal response per unit mass. As shown in this figure, the half cycle when the maximum equivalent displacement ($D_{1U}{}^*{}_{\max}$) occurs corresponds to the half cycle when the maximum momentary energy input occurs.

The equivalent velocity of the maximum momentary input energy of the first modal response ($V_{\Delta E1U}{}^*$) and the response period of the first modal response ($T_1'$) were calculated such that:

$$V_{\Delta E1U}{}^* = \sqrt{2\Delta E_{1U}{}^*{}_{\max}/M_{1U}{}^*}, \tag{26}$$

$$T_1' = 2\Delta t. \tag{27}$$

*5.2. Relationship between the Peak Equivalent Displacement and the Maximum Momentary Input Energy of the First Modal Response*

Figures 33 and 34 show comparisons between the $V_{\Delta E\mu1U}{}^*$–$D_{1U}{}^*$ curve and the $V_{\Delta E1U}{}^*$–$D_{1U}{}^*{}_{\max}$ relationship obtained from the nonlinear time-history analysis, with

the results of Models Tf34 and Tf44 shown in Figures 33 and 34, respectively. These figures confirm that the $V_{\Delta E\mu 1U}{}^{*}-D_{1U}{}^{*}$ curve fits the plots obtained from the time-history analysis results very well. This is because the contribution of the first modal response to the whole response may be large: The effective modal mass ratio of the first mode with respect to the U-axis is large (more than 0.7) in both Models Tf34 and Tf44.

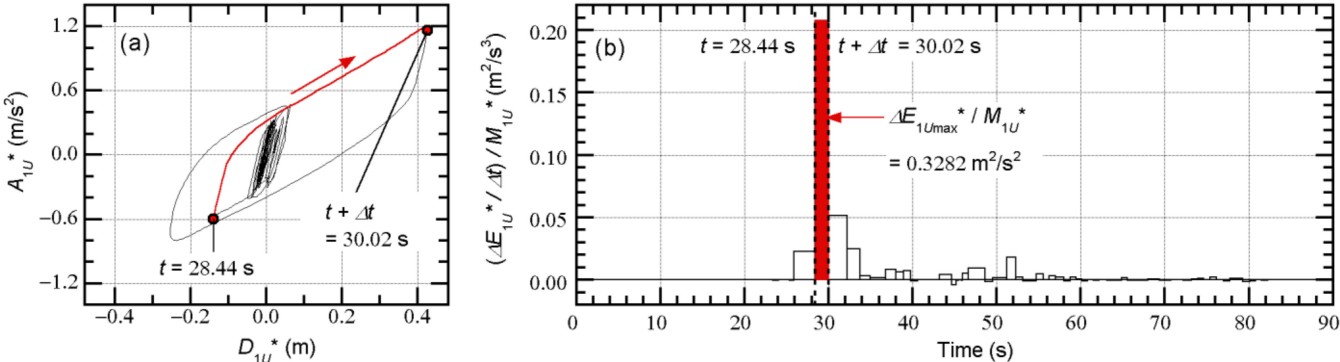

**Figure 32.** Definition of the maximum momentary input energy of the first modal response per unit mass (structural model: Model-Tf44, ground motion: TCU, angle of incidence of seismic input: $\psi = -30°$): (**a**) Hysteresis of the first modal response and (**b**) time-history of the momentary energy input.

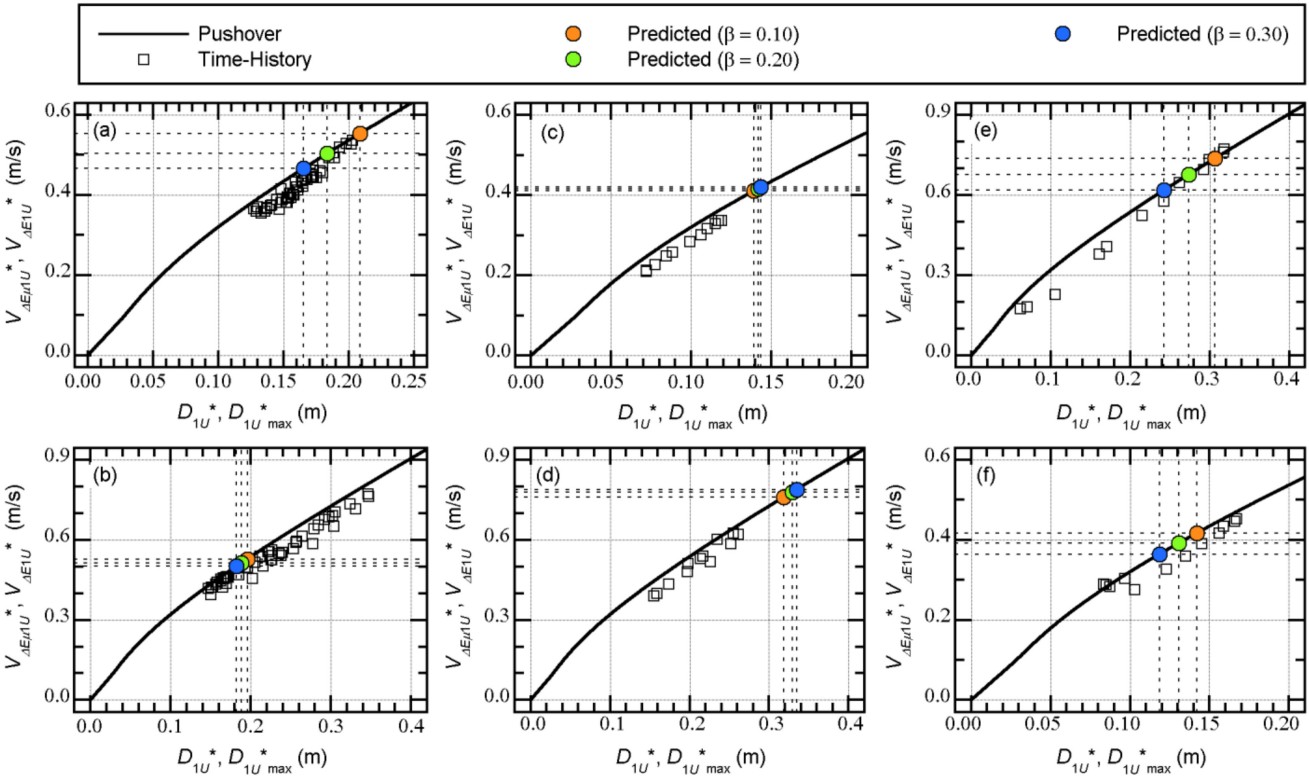

**Figure 33.** Comparison between the $V_{\Delta E\mu 1U}{}^{*}-D_{1U}{}^{*}$ curve and the $V_{\Delta E1U}{}^{*}-D_{1U}{}^{*}{}_{max}$ relationship obtained from the time-history analysis (Model-Tf34): (**a**) Art-1 series; (**b**) Art-2 series; (**c**) UTO0414; (**d**) UTO0416; (**e**) TCU; (**f**) YPT.

Therefore, the accuracy of the predicted $D_{1U}{}^{*}{}_{max}$ relies on the accuracy of the predicted $V_{\Delta E1U}{}^{*}$, which is discussed in the next subsection.

### 5.3. Comparison of the Maximum Momentary Input Energy and the Bidirectional Momentary Input Energy Spectrum

Figure 35 shows the prediction of $V_{\Delta E1U}^{*}$ from the bidirectional $V_{\Delta E}$ spectrum for Model-Tf34. The plots shown in this figure indicate the nonlinear time-history analysis results.

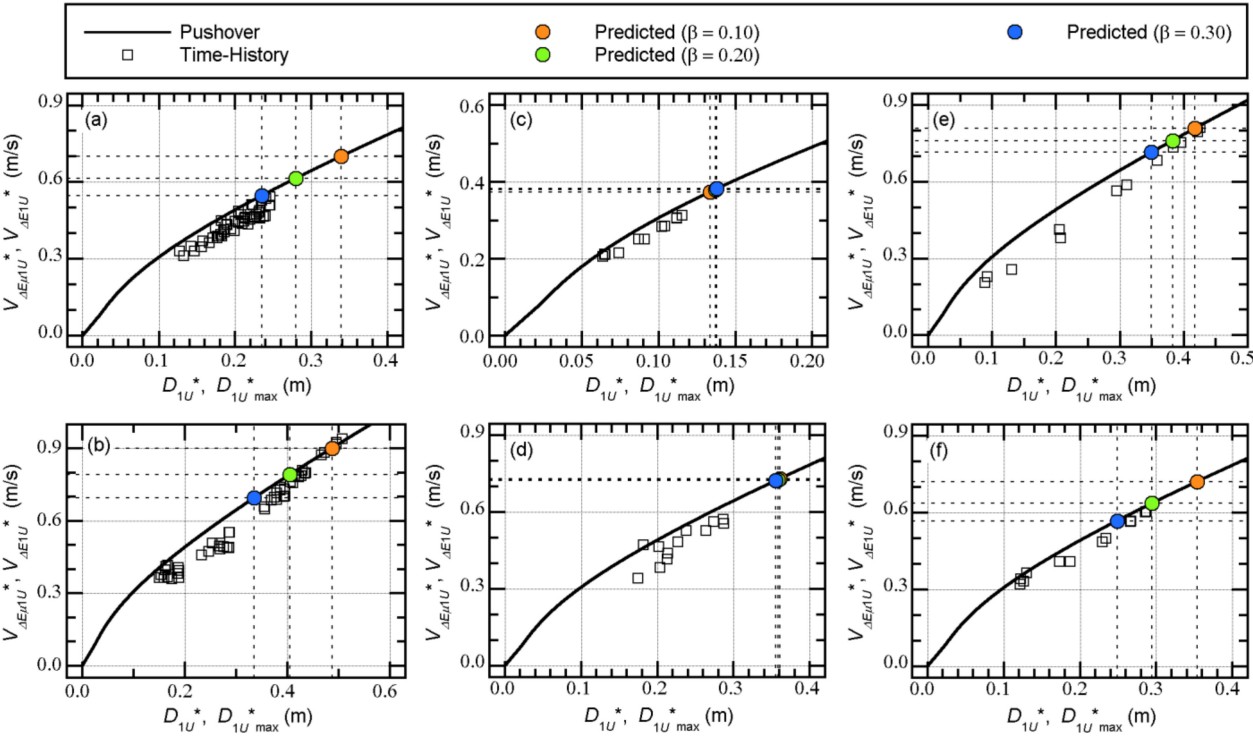

**Figure 34.** Comparison between the $V_{\Delta E\mu1U}^{*}$–$D_{1U}^{*}$ curve and the $V_{\Delta E1U}^{*}$–$D_{1U\ max}^{*}$ relationship obtained from the time-history analysis (Model-Tf44): (**a**) Art-1 series; (**b**) Art-2 series; (**c**) UTO0414; (**d**) UTO0416; (**e**) TCU; (**f**) YPT.

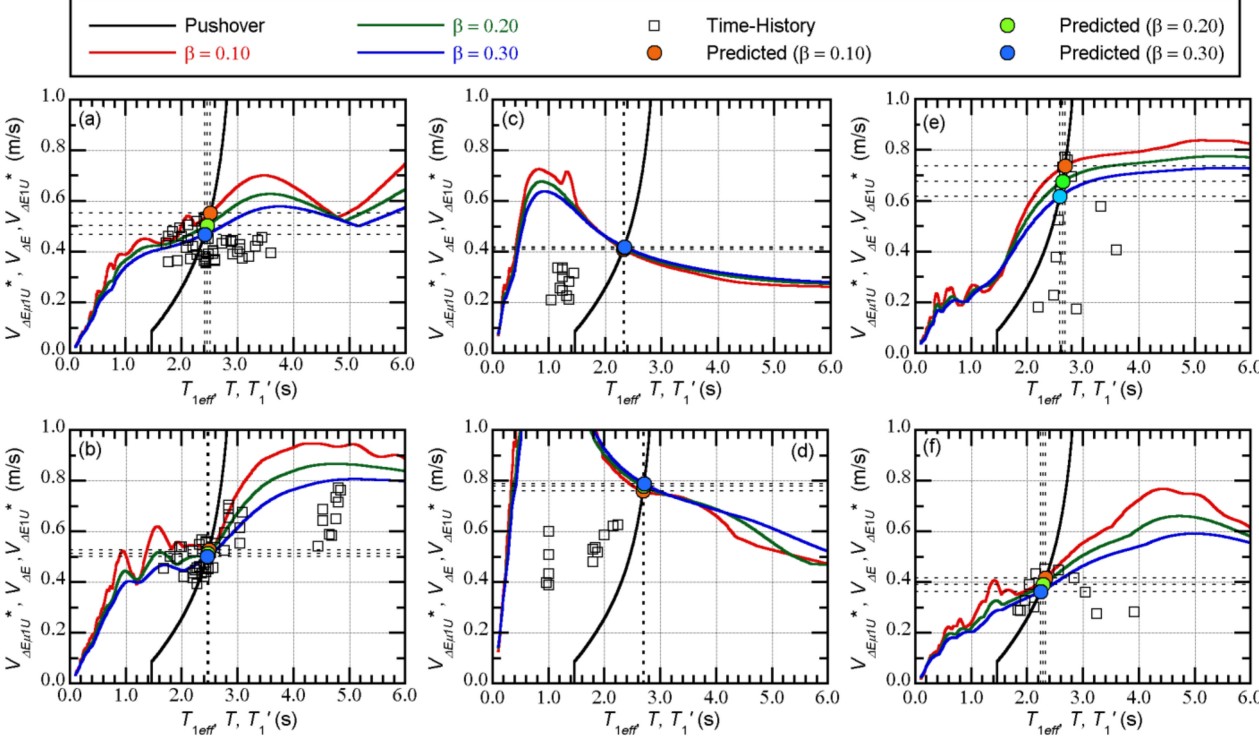

**Figure 35.** Prediction of $V_{\Delta E1U}^{*}$ from the bidirectional $V_{\Delta E}$ spectrum and its accuracy (Model-Tf34): (**a**) Art-1 series; (**b**) Art-2 series; (**c**) UTO0414; (**d**) UTO0416; (**e**) TCU; (**f**) YPT.

In most cases, the bidirectional $V_{\Delta E}$ spectrum with complex damping $\beta = 0.10$ approximated the upper bound of the plot of the time-history analysis results, as shown in Figure 35a,b,e,f. However, in the other cases shown in Figure 35c,d, the plots of the time-history analysis results were below those of the bidirectional $V_{\Delta E}$ spectrum.

From the comparisons between the predicted $V_{\Delta E1U}{}^*$ and that obtained from the time-history analysis results, the predicted $V_{\Delta E1U}{}^*$ provided a conservative estimation, except in the case of the Art-2 series shown in Figure 35b. This is because the predicted response points correspond to the "valley" of the bidirectional $V_{\Delta E}$ spectrum, therefore making the predicted $V_{\Delta E1U}{}^*$ smaller.

Figure 36 shows the prediction of $V_{\Delta E1U}{}^*$ from the bidirectional $V_{\Delta E}$ spectrum for Model-Tf44. Similar observations to those made for Model-Tf34 can be made for Model-Tf44.

Based on the above results, it can be concluded that that bidirectional $V_{\Delta E}$ spectrum can approximate the upper bound of the equivalent velocity of the maximum momentary input energy of the first mode $V_{\Delta E1U}{}^*$. As shown in Equation (25), $\Delta E_{1U}{}^* / M_{1U}{}^*$ was calculated from the ground acceleration component of U-axis, while the contribution of V-axis was none. Therefore, it is easily expected that the upper bound of the maximum momentary input energy of the first mode can be approximated by the bidirectional maximum momentary input energy. Comparisons between the unidirectional and bidirectional $V_{\Delta E}$ spectra can be found in Appendix B. Therefore, the upper bound of $V_{\Delta E1U}{}^*$ can be properly predicted using the bidirectional $V_{\Delta E}$ spectrum.

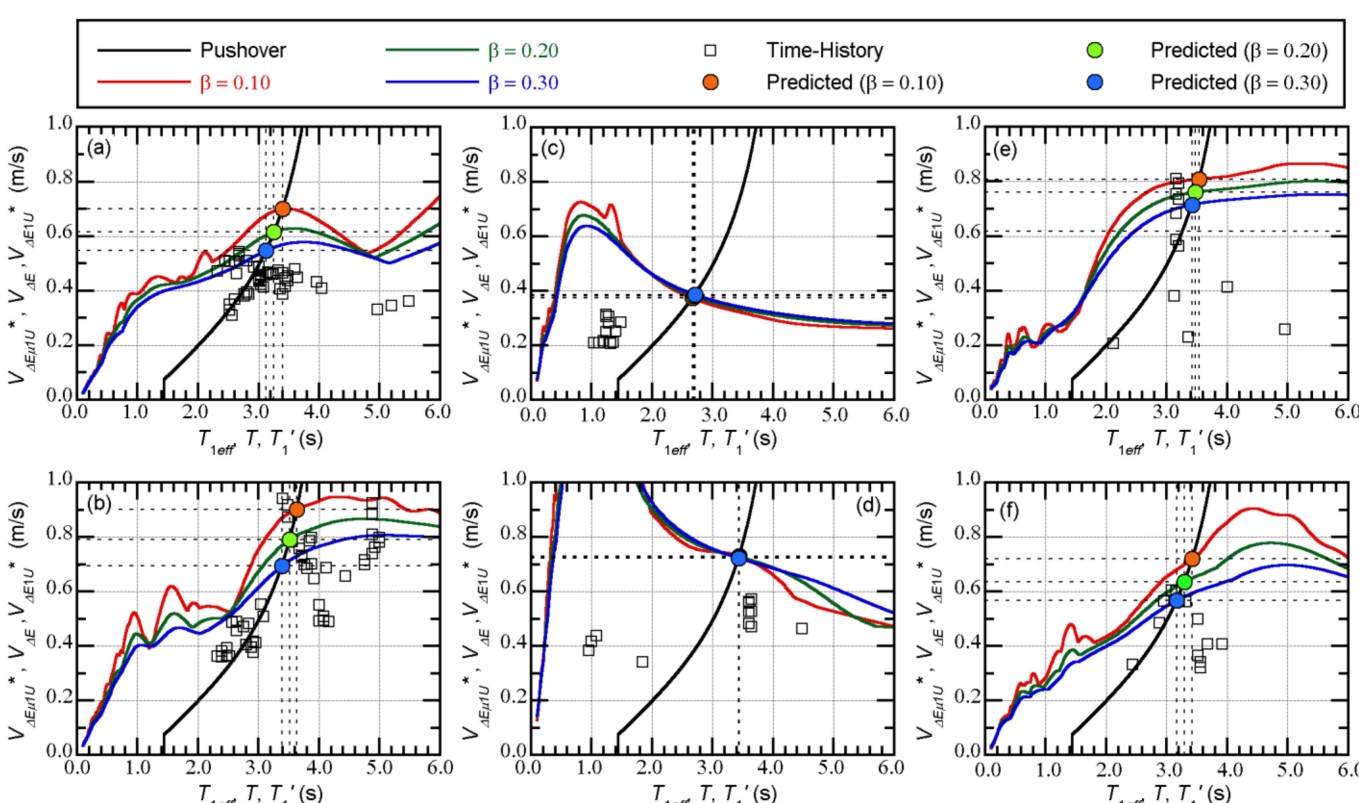

**Figure 36.** Prediction of $V_{\Delta E1U}{}^*$ from the bidirectional $V_{\Delta E}$ spectrum and its accuracy (Model-Tf44): (**a**) Art-1 series; (**b**) Art-2 series; (**c**) UTO0414; (**d**) UTO0416; (**e**) TCU; (**f**) YPT.

### 5.4. Accuracy of the Predicted Peak Equivalent Displacements of the First and Second Modal Responses

Figure 37 shows the accuracy of the predicted peak equivalent displacement of the first and second modal responses, $D_{1U}{}^*{}_{max}$ and $D_{2V}{}^*{}_{max}$, respectively, for Model-Tf34. The predicted peak in the case of complex damping $\beta = 0.10$ approximated the upper bound of

the time-history analysis results, except when the ground motion dataset was the Art-2 series (Figure 37b) or YPT (Figure 37f).

Figure 38 shows the accuracy of the predicted $D_{1U}^{*}{}_{max}$ and $D_{2V}^{*}{}_{max}$ for Model-Tf44. The predicted peak in the case of $\beta = 0.10$ approximated the upper bound of the time-history analysis results in all cases.

It is also observed from Figures 37c–f and 38c–f that the largest $D_{1U}^{*}{}_{max}$ and $D_{2V}^{*}{}_{max}$ did not occur simultaneously. To understand this phenomena, Figures 39 and 40 show $D_{1U}^{*}{}_{max}$ and $D_{2V}^{*}{}_{max}$ for varying angles of incidence of the seismic input ($\psi$) for both models. As shown here, the angle where the largest $D_{2V}^{*}{}_{max}$ occurred was different from the angle where the largest $D_{1U}^{*}{}_{max}$ occurs, with the difference between the two angles being approximately 90°. This was clearly observed in the cases of TCU and YPT for both models.

Based on the above discussion, the upper bound of the peak equivalent displacements of the first and second modal responses, $D_{1U}^{*}{}_{max}$ and $D_{2V}^{*}{}_{max}$, can be predicted by the updated MABPA.

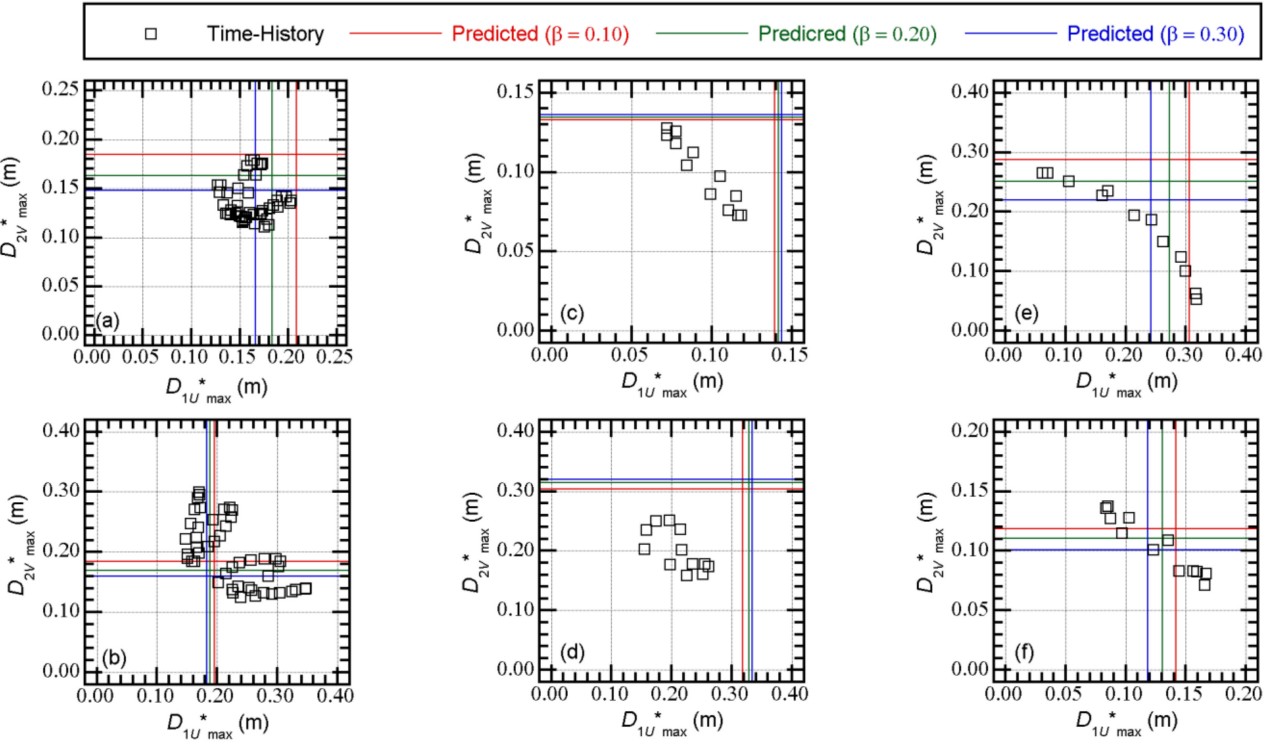

**Figure 37.** Accuracy of the predicted peak equivalent displacements of the first two modes (Model-Tf34): (**a**) Art-1 series; (**b**) Art-2 series; (**c**) UTO0414; (**d**) UTO0416; (**e**) TCU; (**f**) YPT.

The variation in $D_{1U}^{*}{}_{max}$ and $D_{2V}^{*}{}_{max}$ due to the angles of incidence of the seismic input $\psi$ may explain the variation in the peak displacement of the isolatiors shown in Figures 29 and 30. As the example, the response of Model-Tf44 subjected to YPT ground motion was focused on. From Figure 40c, the angle ($\psi$) at which the largest $D_{1U}^{*}{}_{max}$ occurred was $-30°$. While the peak displacement at isolator $X_{1A}Y_6$ shown in Figure 30c, the peak displacement at the angle ($\psi$) of $-30°$ was close to the largest peak value. On the contrary, the peak displacement at isolator $X_{6A}Y_1$ at the angle ($\psi$) of $-30°$ was close to the smallest peak value.

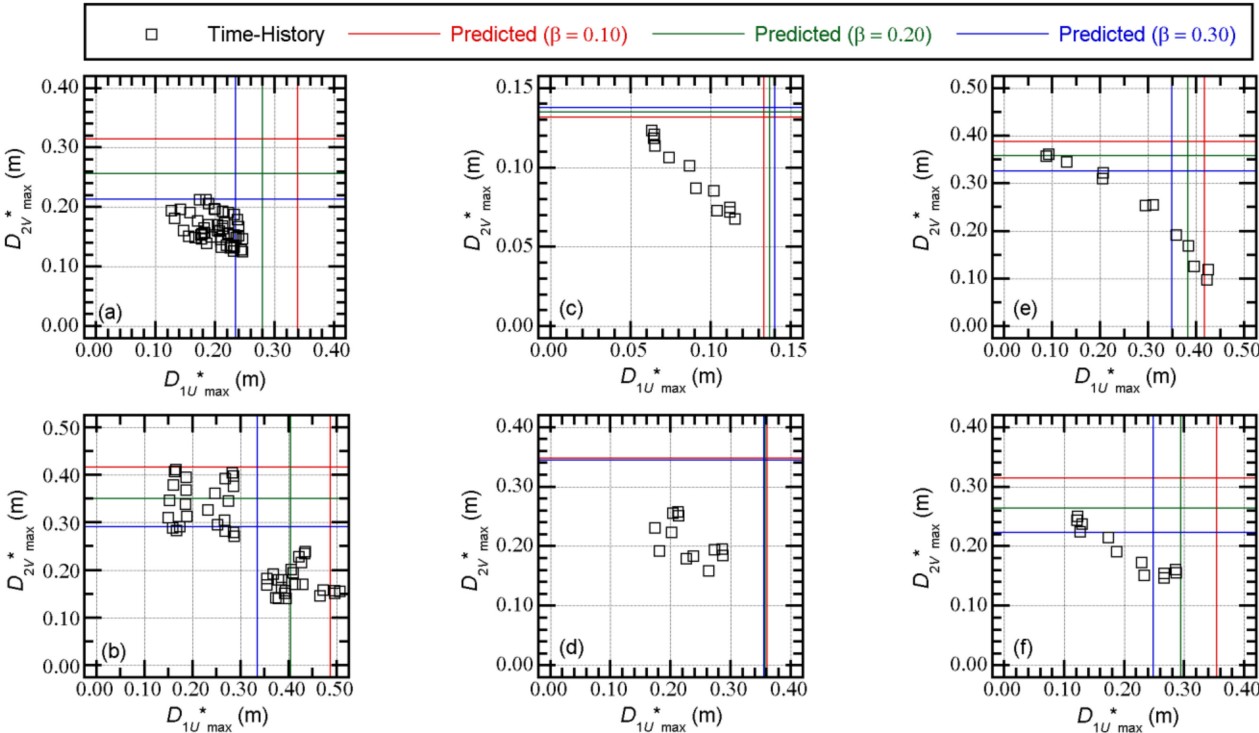

**Figure 38.** Accuracy of the predicted peak equivalent displacements of the first two modes (Model-Tf44): (**a**) Art-1 series; (**b**) Art-2 series; (**c**) UTO0414; (**d**) UTO0416; (**e**) TCU; (**f**) YPT.

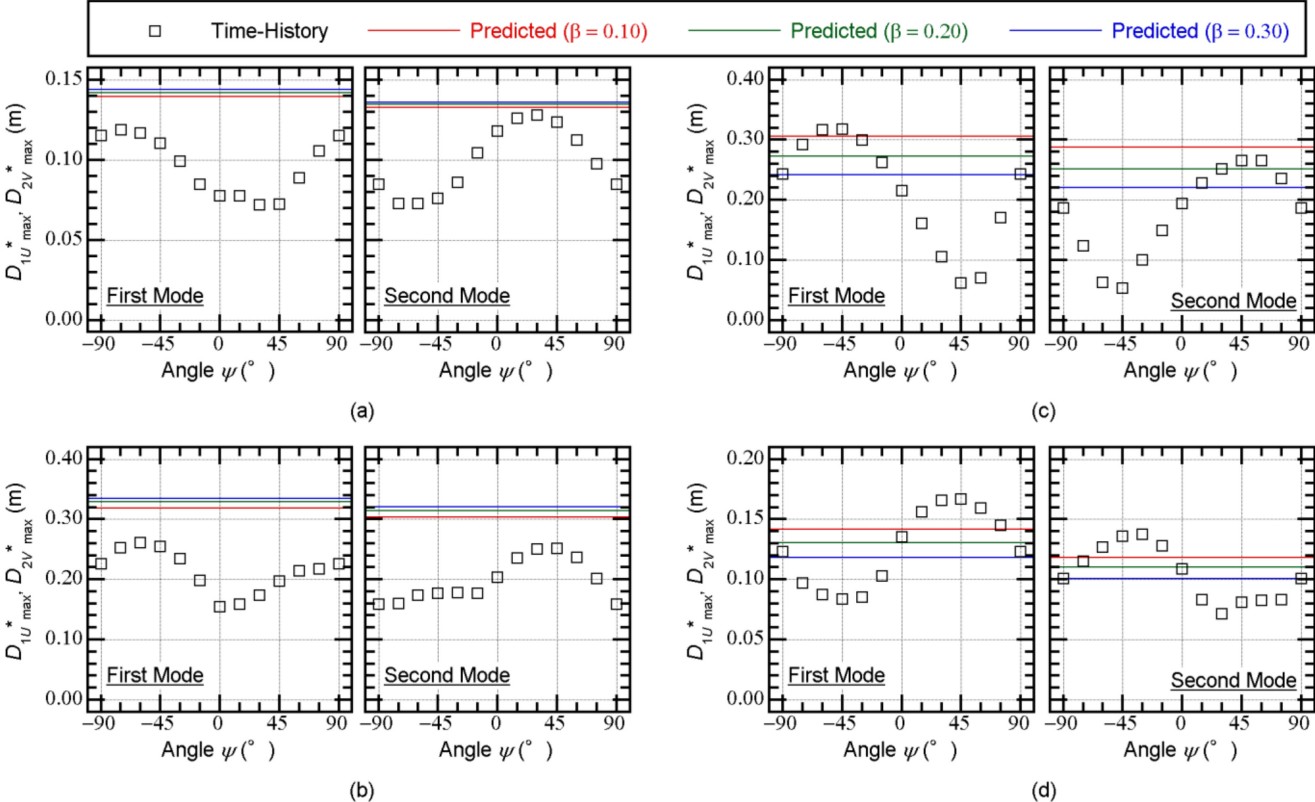

**Figure 39.** Peak equivalent displacements of the first two modes for various directions of seismic input (Model-Tf34): (**a**) UTO0414; (**b**) UTO0416; (**c**) TCU; (**d**) YPT.

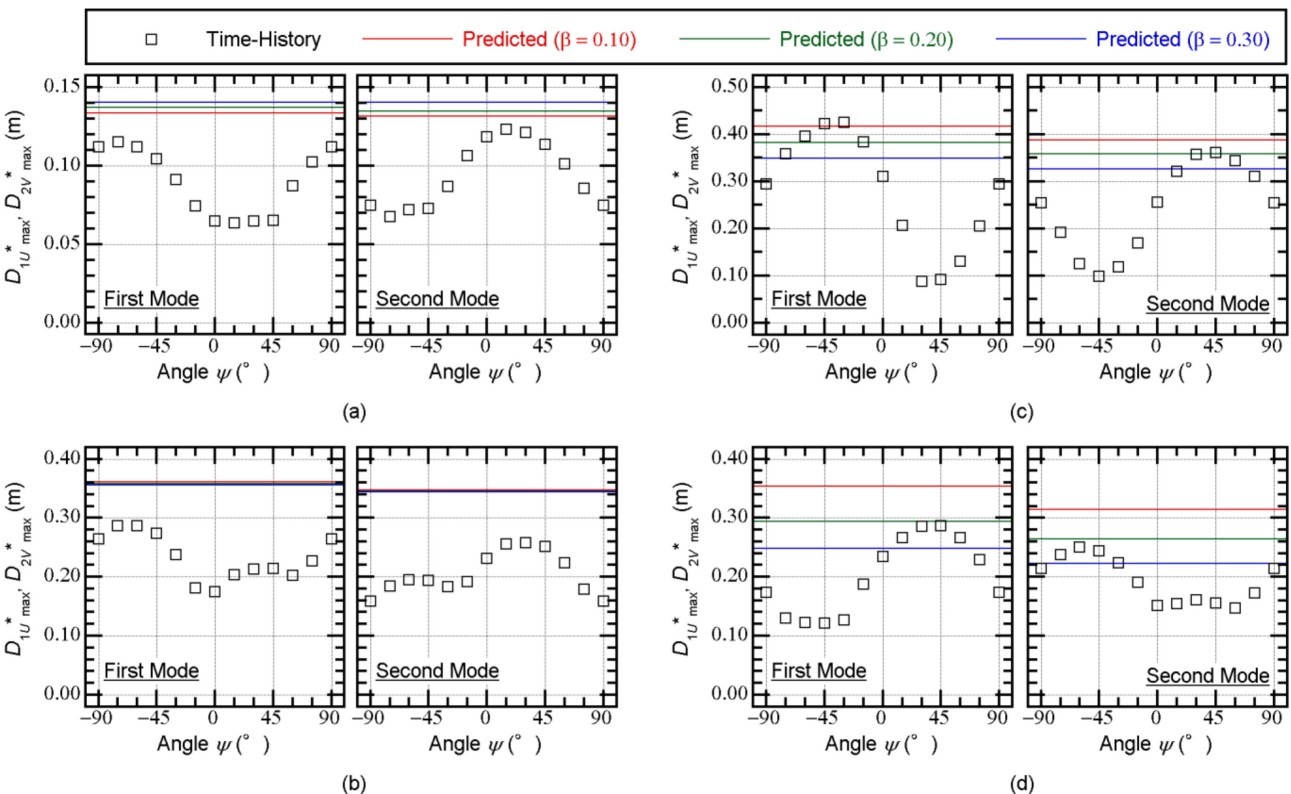

**Figure 40.** Peak equivalent displacements of the first two modes for various directions of seismic input (Model-Tf44): (**a**) UTO0414; (**b**) UTO0416; (**c**) TCU; (**d**) YPT.

### 5.5. Contribution of the Higher Mode to the Displacement Response at the Edge of Level 0

As discussed in Section 4.2.2, the accuracy of the predicted horizontal distribution of the peak response at level 0 depends on the ground motion dataset. According to Model-Tf34, the predicted largest peak displacement in the Y direction occurred at $X_{1A}$ (the flexible side in the Y direction), while the envelope of the nonlinear time-history analysis results indicates that the largest peak occurred at $X_{6A}$ (the stiff side in the Y direction), the opposite side to $X_{1A}$ in the case of UTO0414, as shown in Figure 26a. Conversely, in the case of TCU shown in Figure 26b, the envelope of the time-history analysis results indicates that the largest peak occurred at $X_{1A}$, which is consistent with the predicted results. In this subsection, the modal response at level 0 is calculated and discussed.

From the time-history of the displacement at the center of mass of level 0, the horizontal displacement in the Y direction at point $j$ on level 0 can be calculated as:

$$d_{Y0j}(t) = y_0(t) - L_{X0j}\theta_0(t). \tag{28}$$

In Equation (28), $L_{X0j}$ is the location of point $j$ in the X direction from the center of mass of level 0. Therefore, the modal response of the horizontal displacement at point $j$ can be calculated as:

$$d_{Y0j1}(t) = \Gamma_{1U}\left(\phi_{Y01} - L_{X0j}\phi_{\Theta01}\right)D_{1U}{}^*(t), \tag{29}$$

$$d_{Y0j2}(t) = \Gamma_{2V}\left(\phi_{Y02} - L_{X0j}\phi_{\Theta02}\right)D_{2V}{}^*(t), \tag{30}$$

$$d_{Y0jh}(t) = d_{Y0j}(t) - \left\{d_{Y0j1}(t) + d_{Y0j2}(t)\right\}. \tag{31}$$

Figure 41 shows comparisons of the modal responses at the edge of level 0. The structural model shown in this figure is Model-Tf34, the input ground motion dataset is UTO0414, and the angle of incidence of the seismic input ($\psi$) is $-75°$, the angle at which the largest peak response at $X_{1A}$ occurs. Note that "All modes" is the response originally

obtained from the time-history analysis results ($d_{Y0j}(t)$), "First mode" and "Second mode" are the first and second modal responses ($d_{Y0j1}(t)$ and $d_{Y0j2}(t)$, respectively), and "Higher mode" is the higher (residual) modal response calculated from Equation (31) ($d_{Y0jh}(t)$).

In the response of $X_{1A}$, shown in Figure 41a, the contribution of the higher modal response was non-negligible, even though the contribution of the first modal response was predominant. In addition, the sign of the higher modal response at the time the peak response occurred at $X_{1A}$ was opposite to that of the "All mode" response, with the contribution of the higher mode reducing the peak response at $X_{1A}$.

In the response of $X_{6A}$, shown in Figure 41b, the contribution of the first modal response was negligibly small and those of the second and higher modal responses were noticeable. In addition, the sign of the higher modal response at the time the peak response occurred at $X_{6A}$ was the same as that of the "All mode" response, with the contribution of the higher mode increasing the peak response at $X_{6A}$.

This indicates that the reason why the largest peak response in the envelope of the time-history analysis results occurred at $X_{6A}$ (*not* at $X_{1A}$) in the case of UTO0414 can be explained by the contributions of the higher modal response. In the case of UTO0414, the contribution of the higher modal response was non-negligibly large.

Another comparison is made in Figure 42 for the structural model Model-Tf34, the input ground motion dataset TCU, and an angle of incidence of the seismic input ($\psi$) of 60°, where the angle of the largest peak response at $X_{1A}$ occurs. In the response of $X_{1A}$, shown in Figure 42a, the contribution of the first modal response was predominant, while that of the higher modal response was small. Meanwhile, in the response of $X_{6A}$, shown in Figure 42b, the contribution of the first modal response was negligibly small and those of the second and higher modal responses were noticeable.

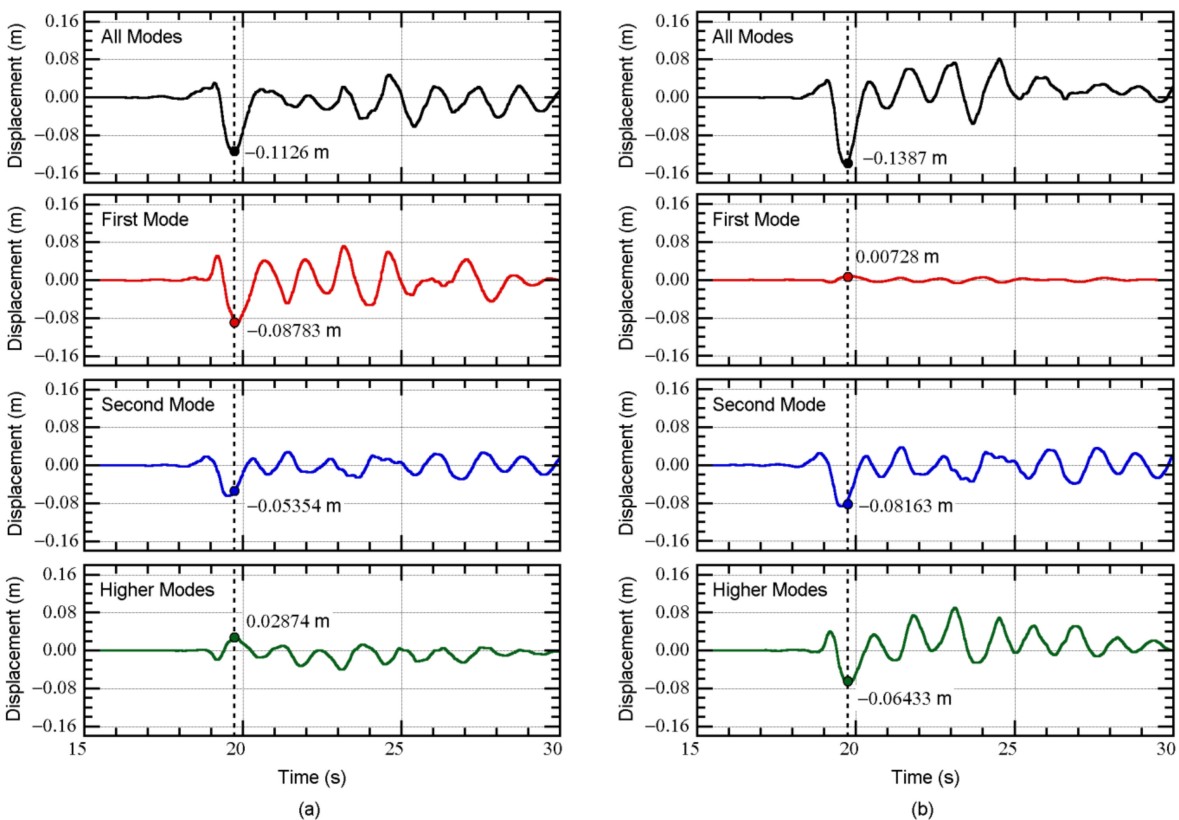

**Figure 41.** Comparisons of the modal responses at the edge of level 0 (structural model: Model-Tf34, ground motion: UTO0414, angle of incidence of seismic input: $\psi = -75°$): (**a**) $X_{1A}$ and (**b**) $X_{6A}$.

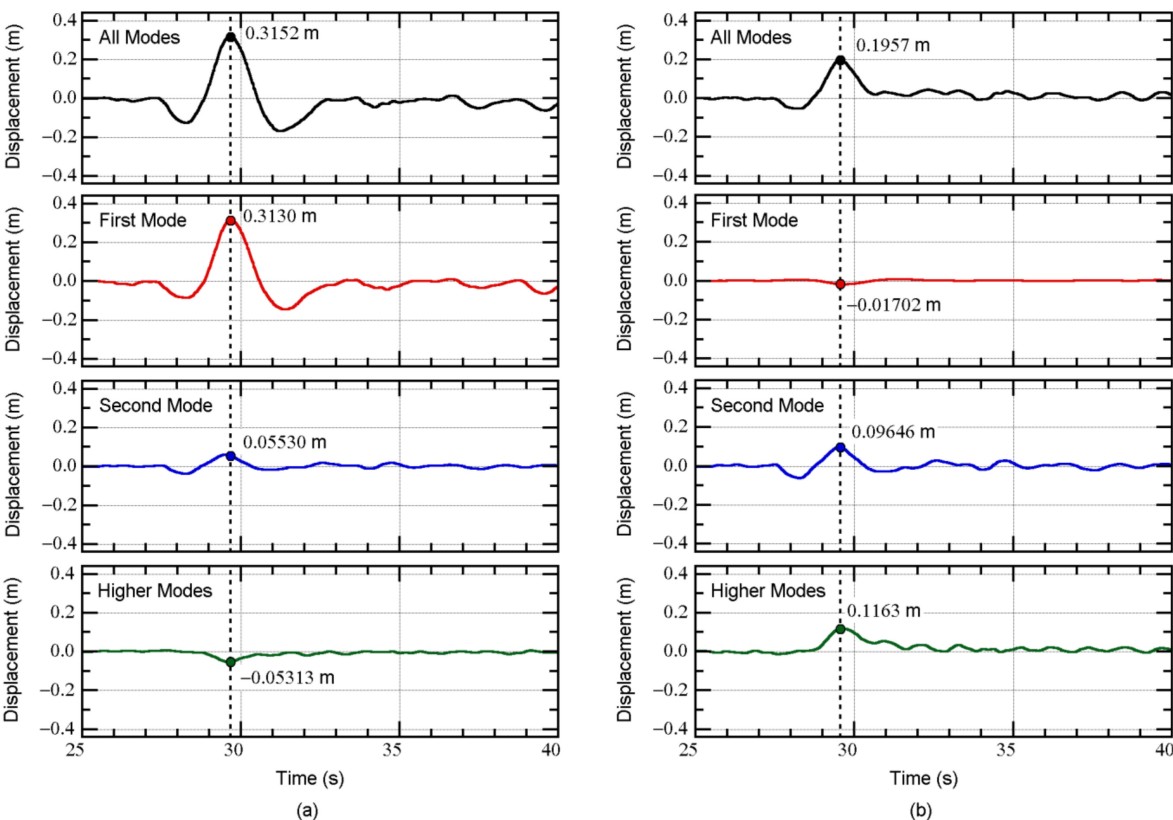

**Figure 42.** Comparisons of the modal responses at the edge of level 0 (structural model: Model-Tf34, ground motion: TCU, angle of incidence of seismic input: $\psi = 60°$): (**a**) $X_{1A}$ and (**b**) $X_{6A}$.

Therefore, the accuracy of the predicted horizontal distribution of the peak response at level 0 relies on the contribution of the higher modal response. The envelope of the time-history analysis results was notably different from the predicted results in the case of UTO0414, because the contribution of the higher modal response was significant. Meanwhile, the predicted results were close to the envelope of the time-history analysis results in the case of TCU because the contribution of the higher modal response was small. To confirm this, Table 6 lists the equivalent velocities of the maximum momentary input energy predicted from the bidirectional $V_{\Delta E}$ spectrum (complex damping ratio ($\beta$) of 0.10) for the two cases. In this table, the values of the first and second modal responses are those predicted using MABPA, while the value of the third mode was predicted assuming that the effective period ($T_{3eff}$) equals the natural period in the elastic range ($T_{3e}$). This table confirms that the contribution of the third mode may be noticeable in the case of UTO0414, while it may be small in the case of TCU.

**Table 6.** Equivalent velocities of the maximum momentary input energy predicted from the bidirectional $V_{\Delta E}$ spectrum ($\beta = 0.100$).

| Ground Motion Set | First Mode $V_{\Delta E}(T_{1eff})$ (m/s) | Second Mode $V_{\Delta E}(T_{2eff})$ (m/s) | Third Mode $V_{\Delta E}(T_{3e})$ (m/s) | Ratio (2nd/1st) | Ratio (3rd/1st) |
|---|---|---|---|---|---|
| UTO0414 | 0.4108 | 0.4232 | 0.7151 | 1.030 | 1.741 |
| TCU | 0.7575 | 0.7319 | 0.2519 | 0.9662 | 0.3325 |

*5.6. Summary of the Discussions*

In this section, the discussions are focused on the four points that are important to discuss the accuracy of MABPA. The summary of the discussions are as follows.

The relationship between the maximum momentary input energy and the peak equivalent displacement of the first modal response was discussed in Section 5.2. The results confirmed that the $V_{\Delta E\mu 1U}{}^*–D_{1U}{}^*$ curve obtained from pushover analysis results fit the plots obtained from the time-history analysis results very well. This is because the contribution of the first modal response to the whole response may be large in both Models Tf34 and Tf44.

Then, the predictability of the maximum momentary input energy of the first modal response was discussed in Section 5.3. The results confirmed that bidirectional $V_{\Delta E}$ spectrum can approximate the upper bound of the equivalent velocity of the maximum momentary input energy of the first mode $V_{\Delta E1U}{}^*$. This is because that the maximum momentary input energy of the first mode was calculated from the unidirectional input in U-axis, while the effect of simultaneous bidirectional input was included in the calculation of bidirectional $V_{\Delta E}$ spectrum automatically.

Since (a) the $V_{\Delta E\mu 1U}{}^*–D_{1U}{}^*$ curve can be properly predicted from the pushover analysis results, and (b) the upper bound of the equivalent velocity of the maximum momentary input energy can be predicted via the bidirectional $V_{\Delta E}$ spectrum, the largest peak equivalent displacement should be predicted accurately. The accuracy of the predicted equivalent displacements of the first and second modal responses was discussed in Section 5.4. The results confirmed that the upper bound of the peak equivalent displacement of the first and second modal responses can be predicted accurately.

Although the upper bound of the peak equivalent displacement of the first and second modal responses can be predicted accurately, the accuracy of the predicted horizontal distributions of the peak displacement at level 0 depends on the ground motion dataset. Section 5.5 discussed the contribution of the higher mode to the displacement response at the edge of level 0. The results showed that higher modal responses may not be negligible for the prediction of the peak displacement at the edge of level 0. In the MABPA prediction, only the contributions of the first and second modal responses were considered. Therefore, a discrepancy of the predicted results from the time-history analysis may occur because of the lack of a contribution from the higher modal responses.

## 6. Conclusions

In this study, the main building of the former Uto City Hall was investigated as a case study of the retrofitting of an irregular reinforced concrete building using the base-isolation technique. The nonlinear peak responses of two retrofitted building models subjected to horizontal bidirectional ground motions were predicted by MABPA, and the accuracy of the method was evaluated. The main conclusions and results are as follows:

- The predicted peak response according to the updated MABPA agreed satisfactorily with the envelope of the time-history analysis results. The peak relative displacement at $X_{3A}Y_3$ at each floor can be satisfactorily predicted. The predicted distribution of the peak displacement at level 0 (just above the isolation layer) approximated the envelope of the nonlinear time-history analysis results, even though in some cases, the predicted distributions differed from the envelope of the nonlinear time-history analysis. A discrepancy between the predicted results and nonlinear time-history analysis may occur because of the lack of a contribution from the higher modal responses.
- The relationship between the equivalent velocity of the maximum momentary input energy of the first modal response ($V_{\Delta E1U}{}^*$) and the peak equivalent displacement of the first modal response ($D_{1U}{}^*{}_{max}$) can be properly evaluated from the pushover analysis results. The plots obtained from the nonlinear time-history analysis results fit the evaluated curve from the pushover analysis results well.
- The upper bound of the peak equivalent displacements of the first two modal responses can be predicted using the bidirectional $V_{\Delta E}$ spectrum [52]. Comparisons between the predicted peak equivalent displacements and those calculated from the nonlinear time-history analysis results showed that the predicted peak approximated

the upper bound of the nonlinear time-history analysis results. The upper bound of $V_{\Delta E1U}{}^*$ can be approximated by the bidirectional $V_{\Delta E}$ spectrum.

Based on the above findings, the updated MABPA appears to predict the peak responses of irregular base-isolated buildings with accuracy. However, MABPA still has two shortcomings. The first shortcoming is the limitation of the applicability of MABPA. As discussed in a previous study [43], the application of MABPA is limited to buildings classified as torsionally stiff buildings. The current (updated) MABPA has the same restriction. This limitation can be avoided if the torsional resistance of the isolation layer is sufficiently provided, as shown in this study. The second shortcoming involves the contributions of the higher modal responses. In the original MABPA for non-isolated buildings, only the first two modes were considered for the prediction. Therefore, the prediction was less accurate for cases when the response in the stiff-side perimeter was larger than that in the flexible-side perimeter. The contributions of the third and higher modal responses need to be investigated.

Another aspect of this study to be emphasized is the application of the bidirectional $V_{\Delta E}$ spectrum for the prediction of the peak response of a base-isolated building. The results shown in this study imply that the bidirectional $V_{\Delta E}$ spectrum [52] is a promising candidate for a seismic intensity parameter for the peak response. As discussed in a previous study [53], one of the biggest advantages of the bidirectional momentary input energy is that it can be directly calculated from the Fourier amplitude and phase angle of the ground motion components using a time-varying function of the momentary energy input, without knowing the time-history of the ground motion. This means that researchers can eliminate otherwise unavoidable fluctuations from the nonlinear time-history analysis results. Therefore, the pushover analysis and the bidirectional $V_{\Delta E}$ spectrum are an optimal combination to understand the fundamental characteristics of both base-isolated and non-isolated asymmetric buildings.

The optimal distribution of the hysteresis dampers according to the design of the isolation layer needed to minimize the torsional response was not discussed in this study. However, the updated MABPA presented here can help in the optimization of the damper distribution. The next update of MABPA for base-isolated irregular buildings with other kinds of dampers (e.g., linear and nonlinear oil dampers, viscous mass dampers, and other kings of "smart passive dampers" [19]) is also an important issue that will be investigated in subsequent studies.

**Author Contributions:** Conceptualization, K.F. and T.M.; data curation, K.F. and T.M.; formal analysis, K.F. and T.M.; funding acquisition, K.F.; investigation, K.F. and T.M.; methodology, K.F.; project administration, K.F.; resources, K.F. and T.M.; software, K.F.; supervision, K.F.; validation, K.F.; visualization, K.F.; writing—original draft, K.F.; writing—review and editing, K.F. All authors have read and agreed to the published version of the manuscript.

**Funding:** This research received no external funding.

**Institutional Review Board Statement:** Not applicable.

**Informed Consent Statement:** Not applicable.

**Data Availability Statement:** The data presented in this study are available on request from the corresponding author. The data are not publicly available because they are not part of ongoing research.

**Acknowledgments:** The ground motions used in this study were obtained from the website of the National Research Institute for Earth Science and Disaster Resilience (NIED) (http://www.kyoshin.bosai.go.jp/kyoshin/, last accessed on 14 December 2019) and the Pacific Earthquake Engineering Research Center (PEER) (https://ngawest2.berkeley.edu/, last accessed on 14 December 2019). The contributions during the beginning stage of this study made by Ami Obikata, a former undergraduate student at the Chiba Institute of Technology, are greatly appreciated.

**Conflicts of Interest:** The authors declare no conflict of interest.

## Appendix A. Time-Histories of the Recorded Ground Motions Used in This Study

Table A1 shows the date of event, magnitude (Meteorological Agency Magnitude $M_J$, or moment magnitude $M_W$), location of the epicenter, distance, and station name of each record. Figures A1–A4 show the time-histories and orbits of the original ground motion records.

**Table A1.** Event date, magnitude, location of the epicenter, distance, and station name of each record.

| ID | Event Date | Magnitude | Distance | Station Name | Direction of Components | |
|---|---|---|---|---|---|---|
| | | | | | $\xi$-Dir | $\zeta$-Dir |
| UTO0414 | 14 April 2016 | $M_J = 6.5$ | 15 km | K-Net UTO (KMM008) | EW | NS |
| UTO0416 | 16 April 2016 | $M_J = 7.3$ | 12 km | K-Net UTO (KMM008) | EW | NS |
| TCU | 20 September 1999 | $M_W = 7.6$ | 0.89 km * | TCU075 | Major ** | Minor ** |
| YPT | 17 August 1999 | $M_W = 7.5$ | 4.83 km * | Yarimca | Major ** | Minor ** |

* This distance is the closest distance from the rupture plane defined in the Pacific Earthquake Engineering Research Center (PEER) database, while those from the Japanese database are the epicentral distances. ** Horizontal major and minor axes were determined following the works of Arias [63] and Penzien and Watabe [64].

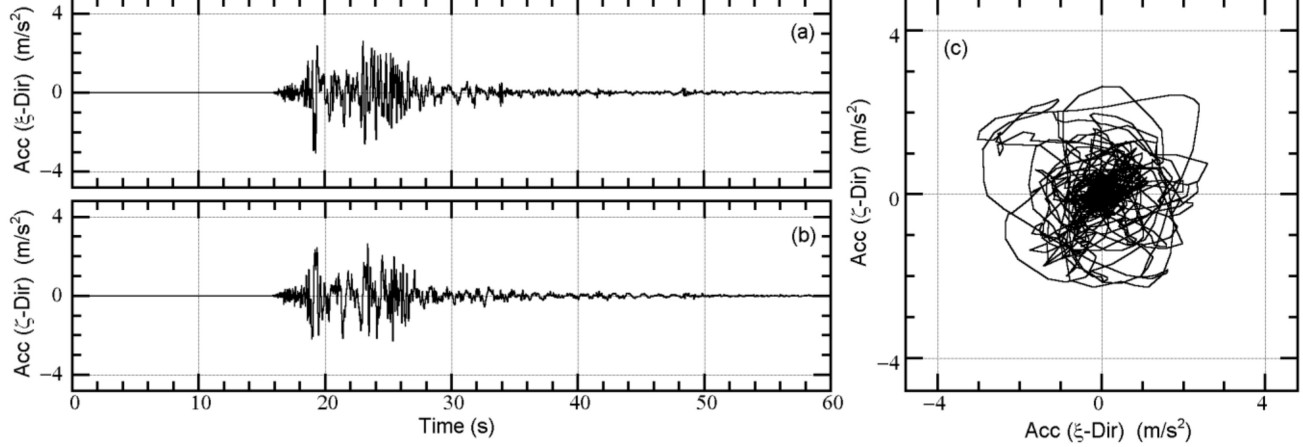

**Figure A1.** Two components of the recorded ground motion (UTO0414): (**a**) $\xi$ direction; (**b**) $\zeta$ direction; (**c**) orbit.

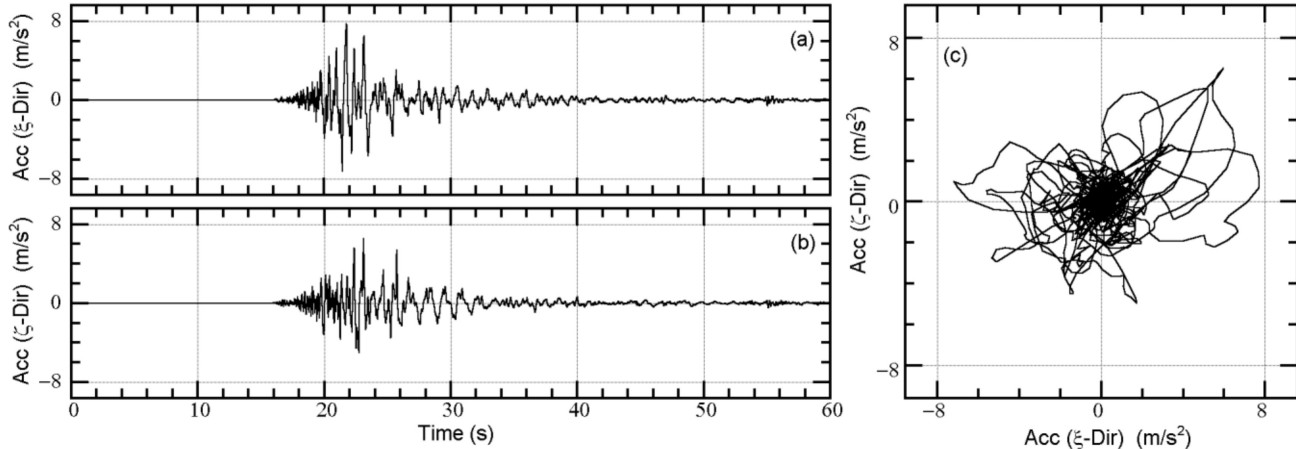

**Figure A2.** Two components of the recorded ground motion (UTO0416): (**a**) $\xi$ direction; (**b**) $\zeta$ direction; (**c**) orbit.

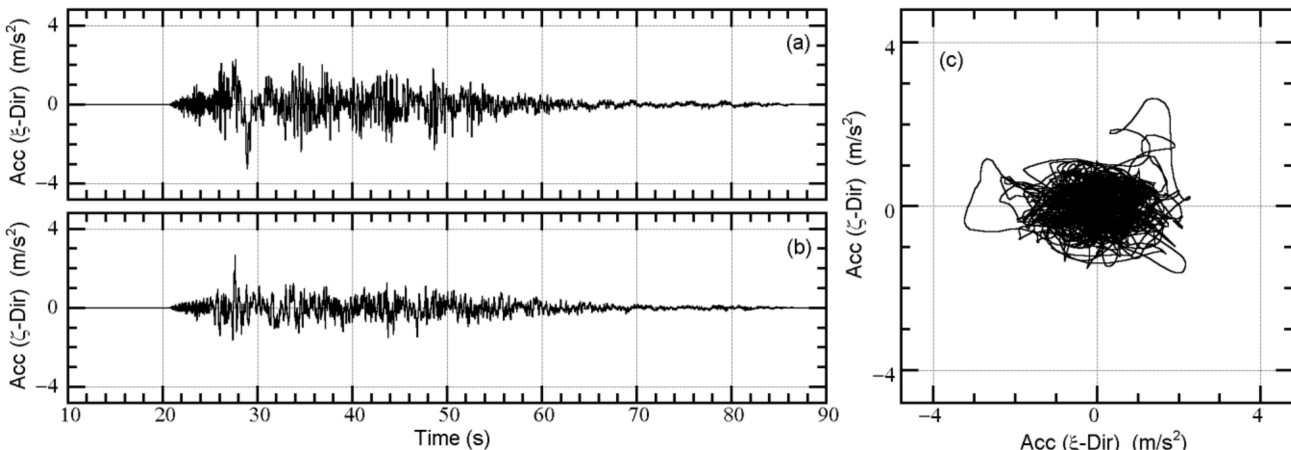

**Figure A3.** Two components of the recorded ground motion (TCU): (**a**) $\xi$ direction; (**b**) $\zeta$ direction; (**c**) orbit.

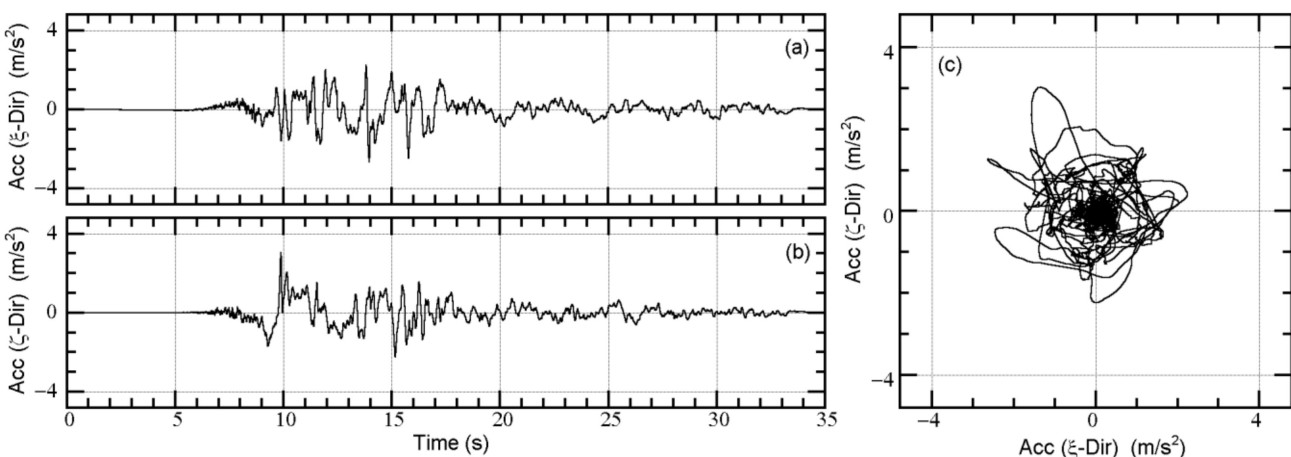

**Figure A4.** Two components of the recorded ground motion (YPT): (**a**) $\xi$ direction; (**b**) $\zeta$ direction; (**c**) orbit.

## Appendix B. Comparisons of the Unidirectional and Bidirectional $V_{\Delta E}$ Spectra

Figure A5 shows the comparisons of the unidirectional and bidirectional $V_{\Delta E}$ spectra for Art-1 and Art-2 series. In this figure, the unidirectional $V_{\Delta E}$ spectrum was calculated as the maximum obtained from the linear time-history analysis using all possible rotations between $0°$ and $360°$ degrees with intervals of $5°$. Viscous damping (damping ratio 0.10) was considered, while the bidirectional $V_{\Delta E}$ spectrum was calculated using the time-varying function described in Section 2.2.1. Two damping models, complex and viscous damping (damping ratio 0.10), were considered. Note again that the bidirectional $V_{\Delta E}$ spectrum calculated from time-varying function is independent of phase-shift. As shown in this figure, the bidirectional $V_{\Delta E}$ spectrum approximates the maximum of the unidirectional $V_{\Delta E}$ spectrum.

Figure A6 shows comparisons of the unidirectional and bidirectional $V_{\Delta E}$ spectra for recorded ground motion datasets. In this figure, the maximum, minimum, and medium of the set of geometrical means obtained using all possible rotations between $0°$ and $90°$ (GMRotD50) defined by Boore et al. [65] of unidirectional $V_{\Delta E}$ spectra are shown. As shown in this figure, the bidirectional $V_{\Delta E}$ spectrum approximates the maximum of the unidirectional $V_{\Delta E}$ spectrum.

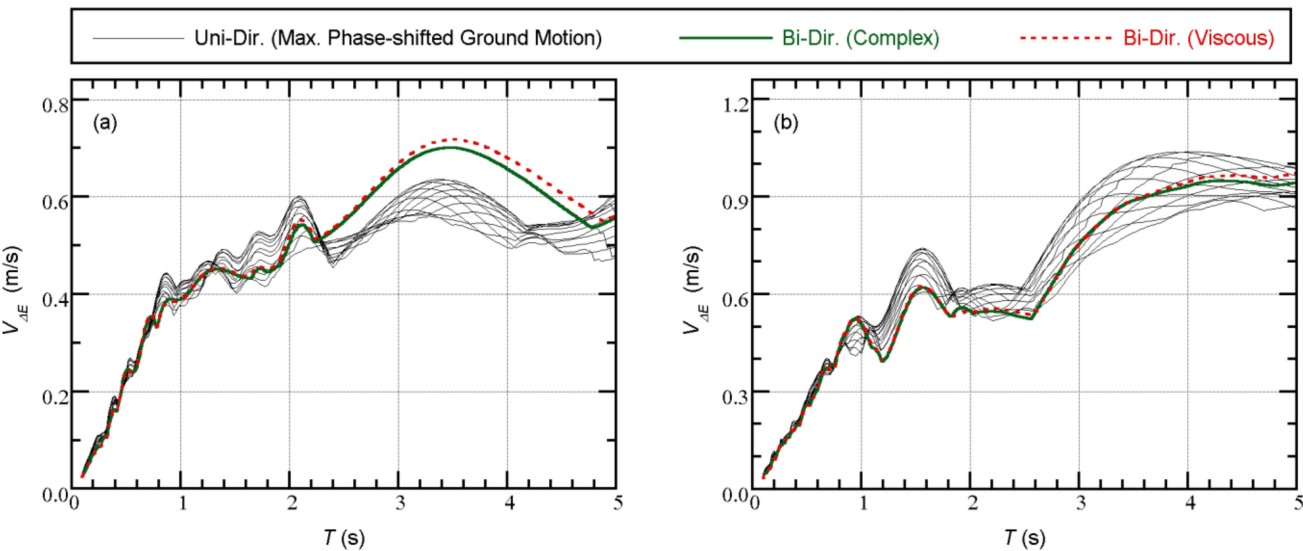

**Figure A5.** Comparisons of the unidirectional and bidirectional $V_{\Delta E}$ spectra for artificial ground motion datasets: (**a**) Art-1 series and (**b**) Art-2 series.

**Figure A6.** Comparisons of the unidirectional and bidirectional $V_{\Delta E}$ spectra for recorded ground motion datasets: (**a**) UTO0414; (**b**) UTO0416; (**c**) TCU; (**d**) YPT.

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
