# Peer review of "Application of Mode-Adaptive Bidirectional Pushover Analysis to an Irregular Reinforced Concrete Building Retrofitted via Base Isolation"

_applsci, doi:10.3390/app11219829_

Round 1

Reviewer 1 Report

The authors have applied mode-adaptive bidirectional pushover analysis for evaluating the response of the irregular reinforced concrete building. The presented paper is not acceptable for publication in the current form.

  1. The abstract of the paper should be revised so that it can be indicative of the comparison of results, made with the nonlinear time history analysis during the whole study.
  2. Since the topic is not focused and the study phase has too many variables, the literature review missed out on many important aspects of base isolation(doi.org/10.3390/app11062876), irregularities (doi.org/10.1680/jstbu.21.00011) and type of loading (doi.org/10.1016/j.prostr.2019.08.145). 
  3. Despite the irregularity in the plan, the modal analysis results for both retrofitted models elaborate that the first two modes are predominantly translational and the third one is rotational. This behavior portrays that there is not much eccentricity between the loading center and center of mass. It shall be of significance if the further application of the proposed procedure can be extended to buildings having a different kind of behavior.
  4. In section 2, it shall be very helpful if authors can provide an actual photograph of the building in the article, if possible.
  5. Section 2.4 can be included after section 4 as a separate section, before establishing the comparison of results with time history analysis as the primary focus of the paper has been the application of the nonlinear static procedure.
  6. In section 5, although the discreet discussion has been provided under relevant sections, the discussions are primarily portraying the obtained results instead of the reasons. Some elaborative discussion about relevant reasons for obtained results should be added under each subsection of section 5 to enhance the understanding of results for the readers.
  7. In the conclusion section, on page 36, lines 765-66, the reason for the difference of predicted distributions from the envelope should be included in a brief way.
  8. Furthermore, the references from the conclusion section should be eliminated and sentences should be phrased accordingly.

Reviewer 2 Report

I would like to congratulate the authors for their effort to deal with a such an interesting topic. Their work is scientifically sound, well structured and worths publishing. 

Within the text, I do not see any reference to the soil properties (e.g. Vs profile) of the site where the building is located. Please provide them if they are available. Also, foundation compliance and/or kinematic SSI has not been included in the analysis models. Please mention the reasons for that and comment on how your results would be affected if soil structure interaction effects were considered.

Round 2
